# *EgoWorld*:
# TRANSLATING EXOCENTRIC VIEW TO EGOCENTRIC VIEW USING RICH EXOCENTRIC OBSERVATIONS

**Junho Park[1], Andrew Sangwoo Ye[2], Taein Kwon[3†]**
[1]AI Lab, LG Electronics, [2]KAIST, [3]Visual Geometry Group, University of Oxford

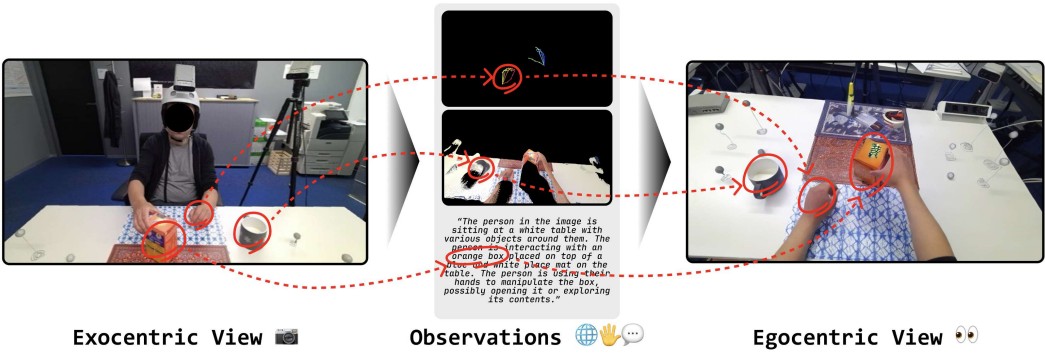

Figure 1: *EgoWorld* translates a single exocentric view into an egocentric view. By leveraging rich multi-modal exocentric observations, such as point clouds, 3D hand poses, and textual descriptions, *EgoWorld* is able to generate high-quality egocentric views, even in unseen scenarios. Each observed modality provides complementary information that contributes to the accurate and realistic reconstruction of the egocentric view.

## ABSTRACT

Egocentric vision is essential for both human and machine visual understanding, particularly in capturing the detailed hand-object interactions needed for manipulation tasks. Translating third-person views into first-person views significantly benefits augmented reality (AR), virtual reality (VR) and robotics applications. However, current exocentric-to-egocentric translation methods are limited by their dependence on 2D cues, synchronized multi-view settings, and unrealistic assumptions such as the necessity of an initial egocentric frame and relative camera poses during inference. To overcome these challenges, we introduce *EgoWorld*, a novel framework that reconstructs an egocentric view from rich exocentric observations, including point clouds, 3D hand poses, and textual descriptions. Our approach reconstructs a point cloud from estimated exocentric depth maps, reprojects it into the egocentric perspective, and then applies diffusion model to produce dense, semantically coherent egocentric images. Evaluated on four datasets (*i.e.,* H2O, TACO, Assembly101, and Ego-Exo4D), *EgoWorld* achieves state-of-the-art performance and demonstrates robust generalization to new objects, actions, scenes, and subjects. Moreover, *EgoWorld* exhibits robustness on in-the-wild examples, underscoring its practical applicability. Project page is available at https://redorangeyellowy.github.io/EgoWorld/.

## 1 INTRODUCTION

Egocentric vision plays a crucial role in advancing visual understanding for both humans and intelligent systems (Ardeshir & Borji, 2018; Grauman et al., 2024; Kwon et al., 2021; Sener et al.,

---

†Corresponding author.

2022). Egocentric views are particularly valuable for capturing detailed hand-object interactions, which are essential in skill-intensive tasks such as cooking, assembling, or playing instruments. However, most existing resources are recorded from third-person perspectives, primarily due to the limited availability of head-mounted cameras and wearable recording devices. Consequently, the ability to generate or predict egocentric images from exocentric inputs holds significant promise for enhancing instructional videos and applications in augmented reality (AR), virtual reality (VR), and robotics, where perception is inherently egocentric. For example, instructional videos are often recorded from a third-person viewpoint, which can be challenging for viewers to follow due to the mismatched perspectives. Translating these videos into a first-person view enables more intuitive guidance by clearly showing detailed finger placements during a task. Moreover, this translation capability unlocks the development of robust, user-centered world models (Wong et al., 2022; Chen et al., 2023; Gao et al., 2023) that capture the spatial and temporal details necessary for real-time perception, planning, and interaction at scale.

Although exocentric-to-egocentric view translation holds great promise, it remains a particularly difficult challenge in computer vision. The main obstacle stems from the substantial visual and geometric differences between third-person and first-person views. Egocentric views focus on hands and objects with the fine detail necessary for precise manipulation, whereas exocentric views offer a wider context and kinematic cues but lack emphasis on these intricate interactions. Bridging these views is fundamentally under-constrained and cannot be addressed by geometric alignment alone, due to factors such as occlusions, restricted fields of view, and appearance changes across different viewpoints. For instance, elements like the inner pages of a book may be completely obscured in an exocentric perspective but still need to be realistically inferred in the egocentric output. Moreover, reconstructing background details in the egocentric view, which are invisible from the exocentric perspective, is a nontrivial task.

Recently, the impressive achievements of diffusion models (Rombach et al., 2022; Ho et al., 2020) have opened up new possibilities for applying generative techniques to the task of exocentric-to-egocentric view translation. However, many existing approaches rely on restrictive input conditions, such as multi-view images (Liu et al., 2024a), known relative camera pose (Cheng et al., 2024), or a reference egocentric frame to generate subsequent ones (Xu et al., 2025), making them impractical for scenarios where only single view images are available. More closely, Exo2Ego (Luo et al., 2024b) attempts to generate egocentric views from a single exocentric image. Yet, it depends heavily on accurate 2D hand layout predictions for structure transformation, which can be unreliable in cases of occlusion, viewpoint ambiguity, or cluttered environments. Furthermore, it struggles to generalize to novel environments and objects, often overfitting to the training dataset. Overall, current methods lack the detailed understanding of exocentric observations necessary to synthesize precise and realistic hand-object interactions from a first-person view.

To address the limitations of current approaches, we propose *EgoWorld*, a novel framework for translating exocentric views into egocentric views using rich exocentric observations, as illustrated in Fig. 1. Our method employs a two-stage pipeline to reconstruct the egocentric view: (1) extracting diverse observations from the exocentric view, including projected point clouds, 3D hand poses, and textual descriptions; and (2) reconstructing the egocentric view based on these extracted cues. In the first stage, we construct a point cloud by combining the input exocentric RGB image with the estimated exocentric depth map, which is scale-aligned by the 3D exocentric hand pose for spatial calibration. This point cloud is then transformed into the egocentric view using a transformation matrix computed from the predicted 3D hand poses in both views. After the projection of the point cloud, a sparse egocentric image is obtained and it is subsequently reconstructed into a dense, high-quality egocentric image using a diffusion-based inpainting model. To further enhance the semantic alignment and visual fidelity of the hand-object reconstruction, we incorporate the predicted exocentric text description and estimated egocentric hand pose during the reconstruction process.

We evaluate the effectiveness of *EgoWorld* through extensive experiments conducted on four datasets (*i.e.,* H2O (Kwon et al., 2021), TACO (Liu et al., 2024b), Assembly101 (Sener et al., 2022), and Ego-Exo4D (Grauman et al., 2024)), which provide well-annotated exocentric and egocentric video pairs. Our method achieves state-of-the-art performance on these benchmarks. Owing to its end-to-end design, *EgoWorld* demonstrates strong generalization across various scenarios, including unseen objects, actions, scenes, and subjects. Moreover, evaluations on unlabeled real-world data further confirm its strong in-the-wild generalization ability.

Our main contributions can be summarized as follows:

- We introduce *EgoWorld*, a novel end-to-end framework that reconstructs high-fidelity egocentric views from a single exocentric image by leveraging rich multi-modal cues, including projected point clouds, 3D hand poses, and textual descriptions.

- Our two-stage pipeline uniquely integrates geometric reasoning with semantic information and diffusion-based inpainting model that significantly enhances hand-object interaction fidelity and semantic alignment for generating egocentric images.

- We demonstrate the strong generalization capability of *EgoWorld* through extensive experiments on H2O, TACO, Assembly101, and Ego-Exo4D datasets. Our approach achieves state-of-the-art performance across diverse and previously unseen scenarios (*i.e.,* unseen objects, actions, scenes, and subjects). Additionally, we show *EgoWorld*'s real-world applicability with in-the-wild examples.

## 2 RELATED WORK

### 2.1 EXOCENTRIC-EGOCENTRIC TRANSLATION

Egocentric vision has also been scaling up particularly due to the introduction of benchmarks (Damen et al., 2018; Kwon et al., 2021; Grauman et al., 2022; Damen et al., 2022; Sener et al., 2022; Grauman et al., 2024). Recently, research on exocentric-to-egocentric (and vice versa) translation (Luo et al., 2024a;b; Cheng et al., 2024; Liu et al., 2024a; Xu et al., 2025) has also gained significant attention. Intention-Ego2Exo (Luo et al., 2024a) proposed an intention-driven ego-to-exo video generation framework that leverages head trajectory and action descriptions to guide content-consistent and motion-aware video synthesis. Exo2Ego (Luo et al., 2024b) introduced a two-stage generative framework for exocentric-to-egocentric view translation that leverages structure transformation and diffusion-based hallucination with hand layout priors. 4Diff (Cheng et al., 2024) proposed a 3D-aware diffusion model for translating exocentric images into egocentric views using egocentric point cloud rasterization and 3D-aware rotary cross-attention. Exo2Ego-V (Liu et al., 2024a) presented a diffusion-based method for generating egocentric videos from sparse 360° exocentric views of skilled daily-life activities, addressing challenges like viewpoint variation and motion complexity. EgoExo-Gen (Xu et al., 2025) addressed cross-view video prediction by generating future egocentric frames from an exocentric video, the initial egocentric frame, and textual instructions, using hand-object interaction dynamics as key guidance. However, these works have fatal limitations: dependency of 2D layouts, pre-defined relative camera pose, multi-view or consecutive sequences inputs, and the challenge of integrating multiple external modalities, such as textual description and pose map.

### 2.2 IMAGE COMPLETION

Image completion is a fundamental problem in computer vision, which aims to fill missing regions with plausible contents (Pathak et al., 2016; Liu et al., 2019; Xiong et al., 2019; Song et al., 2018; Zhao et al., 2021; Suvorov et al., 2022; Li et al., 2022). For example, MAT (Li et al., 2022) proposed a transformer-based model for large-hole image inpainting that combines the strengths of transformers and convolutions to efficiently handle high-resolution images. On the other hand, masked image encoding methods learn representations from images corrupted by masking (Vincent et al., 2010; Pathak et al., 2016; Chen et al., 2020; Dosovitskiy et al., 2020; Bao et al., 2021; He et al., 2022). For example, MAE (He et al., 2022) masks random patches of an input image and learns to reconstruct the missing regions. However, these studies have a limitation that they rely solely on the information surrounding the pixels to restore missing area. With the advent of foundational diffusion models (Ho et al., 2020; Song et al., 2020), it has become possible to perform image completion based on various types of conditions. Specifically, latent diffusion model (Rombach et al., 2022) supports flexible conditioning such as text or bounding boxes and enable high-resolution image synthesis, achieving state-of-the-art results in inpainting, class-conditional generation, and other tasks by incorporating cross-attention. Furthermore, the value of diffusion-based models has been demonstrated across a wide range of challenging domains, such as hand-hand or hand-object interaction image generation (Zhang et al., 2024; Park et al., 2024), and motion generation (Cha et al., 2024; Huang et al., 2025).

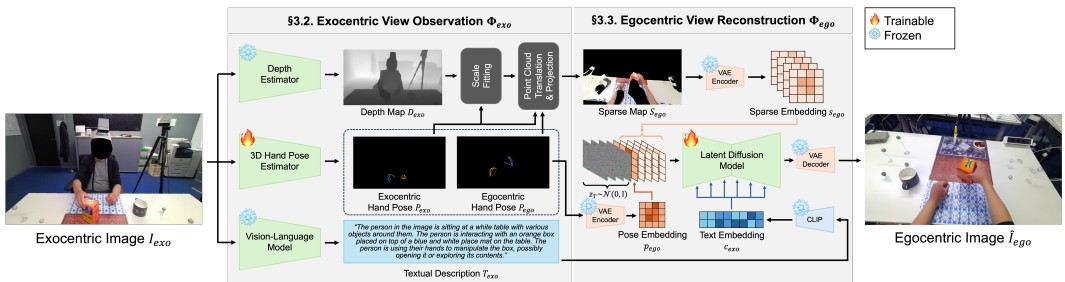

Figure 2: **Overall framework of *EgoWorld*.** *EgoWorld* has a two-stage pipeline : (1) Exocentric view observation $\Phi_{exo}$, which extracts diverse observations from the exocentric view, including projected point clouds, 3D hand poses, and textual descriptions; and (2) egocentric view reconstruction $\Phi_{ego}$, which reconstructs the egocentric view based on cues from the exocentric view observation.

## 3  METHOD

### 3.1  PROBLEM FORMULATION

*EgoWorld* consists of two stages: exocentric view observation $\Phi_{exo}$ and egocentric view reconstruction $\Phi_{ego}$, as shown in Fig. 2. First, given a single exocentric image $I_{exo} \in \mathbb{R}^{H \times W \times 3}$, $\Phi_{exo}$ predicts a corresponding sparse egocentric RGB map $S_{ego} \in \mathbb{R}^{H \times W \times 3}$, 3D egocentric hand pose $P_{ego} \in \mathbb{R}^{N \times 3}$, and a textual description $T_{exo}$. $H$ and $W$ indicates height and width of $I_{exo}$, and $N$ indicates the number of keypoints of the hand. Then, in $\Phi_{ego}$, an egocentric image $\hat{I}_{ego} \in \mathbb{R}^{H \times W \times 3}$ is generated based on the observations predicted in $\Phi_{exo}$. Therefore, *EgoWorld* is formulated as follows:

$$S_{ego}, P_{ego}, T_{exo} = \Phi_{exo}(I_{exo}), \tag{1}$$

$$\hat{I}_{ego} = \Phi_{ego}(S_{ego}, P_{ego}, T_{exo}). \tag{2}$$

### 3.2  EXOCENTRIC VIEW OBSERVATION

Exocentric view observation $\Phi_{exo}$ takes various real-world observations, such as sparse egocentric RGB map $S_{ego}$, 3D egocentric hand pose $P_{ego}$, and textual description $T_{exo}$, from the single exocentric image $I_{exo}$. These observations are essential for the egocentric view reconstruction $\Phi_{ego}$.

First, with an off-the-shelf depth estimator (Wang et al., 2025), an exocentric depth map $D_{exo} \in \mathbb{R}^{H \times W}$ is extracted from $I_{exo}$. Obtaining $D_{exo}$ is essential, because in $\Phi_{ego}$, the reconstruction process relies on $S_{ego}$, which serves as a crucial hint. Specifically, when pixel information from an exocentric view is transformed into an egocentric view, it provides partial observations of the hand, object, or scene, and this serves as a strong basis for approaching the problem from an inpainting perspective.

Next, a 3D exocentric hand pose $P_{exo} \in \mathbb{R}^{N \times 3}$ is extracted from $I_{exo}$ with an off-the-shelf hand pose estimator (Yu et al., 2023). As $D_{exo}$ provides only relative depth and is inherently affected by scale ambiguity, it is crucial to leverage $P_{exo}$ for reasonable scale fitting. Specifically, it is possible to extract a metrically-scaled $P_{exo}$ and an exocentric hand depth map $D_{hand} \in \mathbb{R}^{H \times W}$ from the estimated MANO(Romero et al., 2017)-based mesh of $P_{exo}$. We define a hand region $\Omega_{\text{hand}}$, which is a pixel-level valid area determined by $D_{hand}$, and compute a global scale factor $s^*$ by comparing it with $D_{exo}$ as follows:

$$s^* = \underset{(u,v) \in \Omega_{\text{hand}}}{\text{median}} \frac{D_{hand}(u,v)}{D_{exo}(u,v) + \epsilon}, \tag{3}$$

where $u, v$ indicate the pixel coordinate of depth maps, and $\epsilon$ is a small constant to prevent division by zero. Applying $s^*$ yields a metrically-calibrated exocentric depth map $D'_{exo} = s^* D_{exo}$. Therefore, with $I_{exo}$ and an exocentric camera intrinsic parameter $K_{exo} \in \mathbb{R}^{3 \times 3}$, which is estimated from the off-the-shelf depth estimator, $D'_{exo}$ is utilized to obtain a point cloud $C_{exo} \in \mathbb{R}^{(H \times W) \times 6}$.

To project $C_{exo}$ in the egocentric view, we need an exocentric-to-egocentric view transformation matrix $X \in \mathbb{R}^{4 \times 4}$, which can be computed through a transformation between $P_{exo}$ and $P_{ego}$. However, to the best of our knowledge, there is no model that predicts $P_{ego}$ directly from $I_{exo}$. Thus, we build a powerful-but-simple 3D egocentric hand pose estimator $\phi_{ego}$, which is designed with a simple architecture consisting of a ViT(Dosovitskiy et al., 2020)-based backbone $\phi_{backbone}$ and an MLP-based regressor $\phi_{reg}$. Specifically, after extracting an image feature from $I_{exo}$ with $\phi_{backbone}$, it is fed through $\phi_{reg}$ to obtain $P_{ego}$. We optimize $\phi_{ego}$ with an L2 loss function.

From the obtained $P_{exo}$ and $P_{ego}$, we calculate $X$ between them with the Umeyama algorithm (Umeyama, 1991), which estimates a transformation matrix as follows:

$$X_{ego \rightarrow exo} = (s, \mathbf{R}, \mathbf{t}), \text{ such that } P_{exo} \approx s\mathbf{R}P_{ego} + \mathbf{t}. \tag{4}$$

Here, $s$, $\mathbf{R}$, and $\mathbf{t}$ are the estimated scale, rotation, and translation matrices. Since both $P_{exo}$ and $P_{ego}$ are in metric units, $s$ is expected to be close to 1. The transformation from exocentric to egocentric view is given by $X = (X_{ego \rightarrow exo})^{-1}$. Therefore, we translate $C_{exo}$ with $X$ into $C_{ego}$, project it into egocentric view with an egocentric camera intrinsic parameters $K_{ego} \in \mathbb{R}^{3 \times 3}$, and obtain the sparse egocentric RGB map $S_{ego}$.

Finally, $T_{exo}$ is extracted with an off-the-shelf vision-language model (VLM) (Bai et al., 2023). For example, when $I_{exo}$ and a user-provided question (*i.e., "Describe in detail about the scene and the object that the person is interacting with using their hands."*) are given, VLM outputs the corresponding answer $T_{exo}$. Since $T_{exo}$ contains both the overall contextual information present in the exocentric view and specific details about actions and objects, it significantly aids $\Phi_{ego}$ for reconstructing the faithful egocentric view for unseen scenarios.

### 3.3 Egocentric View Reconstruction

Since $S_{ego}$ only contains partial information observed from the exocentric view, it is necessary to reconstruct the missing regions. Thus, leveraging the powerful latent diffusion model (LDM) (Rombach et al., 2022), we exploit exocentric observations $S_{ego}$, $P_{ego}$, and $T_{exo}$ for $\Phi_{ego}$.

Following the LDM, input images are encoded into the latent embedding using a frozen VAE encoder (Esser et al., 2021), and the denoised latent embedding is decoded into an output image using the frozen VAE decoder. Specifically, we encode $S_{ego}$ to a sparse embedding $s_{ego} \in \mathbb{R}^{64 \times 64 \times 4}$ with VAE encoder. We obtain a 2D egocentric hand pose map $P_{ego}^{2D} \in \mathbb{R}^{512 \times 512 \times 3}$ by projecting $P_{ego}$ with $K_{ego}$, encode $P_{ego}^{2D}$ to 4-channels embedding with VAE encoder, and reduce the number of channels of 4-channels embedding to 1-channel via a channel reduction layer. This layer consists of one convolutional layer, which inputs 4-channel embedding and outputs 1-channel embedding. Therefore, we obtain a 1-channel pose embedding $p_{ego} \in \mathbb{R}^{64 \times 64 \times 1}$.

During training, the ground-truth egocentric image $I_{ego} \in \mathbb{R}^{512 \times 512 \times 3}$ is also encoded to a clean latent $z_0 \in \mathbb{R}^{64 \times 64 \times 4}$ through the VAE encoder, and the noise $\epsilon_t \in \mathbb{R}^{64 \times 64 \times 4}$ is added to $z_0$ to make a noisy embedding $z_t \in \mathbb{R}^{64 \times 64 \times 4}$ with timestep $t$ as follows:

$$z_t = \sqrt{\bar{\alpha}_t} \cdot z_0 + \sqrt{1 - \bar{\alpha}_t} \cdot \epsilon, \epsilon \sim \mathcal{N}(0, \mathbf{I}), \tag{5}$$

where $\bar{\alpha}_t$ denotes the noise level of $t$. By concatenating $s_{ego}$, $p_{ego}$, and $z_t$, we obtain 9-channel latent embedding $z_t' \in \mathbb{R}^{64 \times 64 \times 9}$, which is fed into the input of a pre-trained U-Net. Simultaneously, a textual description $T_{exo}$ is passed through CLIP (Radford et al., 2021) to obtain a text embedding $c_{exo} \in \mathbb{R}^{77 \times 768}$, which serves as guidance for the U-Net of LDM. In this manner, the forward and reverse processes for the denoising network $\epsilon_\theta$ are carried out to predict $\epsilon_t$ with the following objective:

$$\mathcal{L} = \mathbb{E}_{z_0, s_{ego}, p_{ego}, t, c_{exo}, \epsilon_t} \|\epsilon_t - \epsilon_\theta(z_t', t, c_{exo})\|_2^2. \tag{6}$$

During sampling, we start the denoising process from a random Gaussian noise $z_T \sim \mathcal{N}(0, \mathbf{I})$ with well-trained $\epsilon_\theta$. We concatenate $z_T \in \mathbb{R}^{64 \times 64 \times 4}$ with $s_{ego}$ and $p_{ego}$, and feed to $\epsilon_\theta$ to obtain the predicted latent $\hat{z}_0 \in \mathbb{R}^{64 \times 64 \times 4}$ by reversing the schedule in Eq. 5 at each timestep $t \in [1, T]$. We adopt classifier-free guidance (CFG) (Ho & Salimans, 2022) to strengthen textual guidance as follows:

$$\epsilon_t = (1 + w) \cdot \epsilon_\theta(z_t, t, c_{exo}) - w \cdot \epsilon_\theta(z_t, t, \varnothing), \tag{7}$$

where $w$ indicates the scaling factor in CFG, and $\varnothing$ means unconditional. To the end, the final generated egocentric image $\hat{I}_{ego}$ is obtained from $\hat{z}_0$ by passing the VAE decoder.

Table 1: **Comparisons with state-of-the-arts on unseen scenarios (*i.e.,* objects, actions, scenes, and subjects) in H2O (Kwon et al., 2021).** Compared to state-of-the-arts (*i.e.,* pix2pixHD (Wang et al., 2018), pixelNeRF (Yu et al., 2021), and CFLD (Lu et al., 2024)), *EgoWorld* outperforms for all unseen scenarios in all metrics (*i.e.,* FID, PSNR, SSIM, LPIPS, PA-MPJPE, and CLIPScore).

| Scenarios | Unseen Objects | | | | | | Unseen Actions | | | | | |
|---|---|---|---|---|---|---|---|---|---|---|---|---|
| Methods | FID↓ | PSNR↑ | SSIM↑ | LPIPS↓ | PA-MPJPE↓ | CLIPScore↑ | FID↓ | PSNR↑ | SSIM↑ | LPIPS↓ | PA-MPJPE↓ | CLIPScore↑ |
| pix2pixHD (Wang et al., 2018) | 436.25 | 25.012 | 0.2993 | 0.6057 | 18.007 | 0.2302 | 211.10 | 24.420 | 0.2854 | 0.6127 | 17.754 | 0.2450 |
| pixelNeRF (Yu et al., 2021) | 498.23 | 26.557 | 0.3887 | 0.5372 | 15.746 | 0.2270 | 251.76 | 27.061 | 0.3950 | 0.8159 | 14.636 | 0.2315 |
| CFLD (Lu et al., 2024) | 59.615 | 25.922 | 0.4307 | 0.4539 | 7.9971 | 0.2656 | 50.953 | 28.529 | 0.4324 | 0.4593 | 8.1199 | 0.2699 |
| *EgoWorld* (Ours) | **41.334** | **31.171** | **0.4814** | **0.3476** | **7.3178** | **0.2731** | **33.284** | **31.620** | **0.4566** | **0.3780** | **7.2602** | **0.2824** |
| Scenarios | Unseen Scenes | | | | | | Unseen Subjects | | | | | |
| Methods | FID↓ | PSNR↑ | SSIM↑ | LPIPS↓ | PA-MPJPE↓ | CLIPScore↑ | FID↓ | PSNR↑ | SSIM↑ | LPIPS↓ | PA-MPJPE↓ | CLIPScore↑ |
| pix2pixHD (Wang et al., 2018) | 490.32 | 18.567 | 0.2425 | 0.7290 | 20.229 | 0.2159 | 452.13 | 18.172 | 0.3310 | 0.7234 | 21.357 | 0.2311 |
| pixelNeRF (Yu et al., 2021) | 489.13 | 26.537 | 0.2574 | 0.7143 | 17.085 | 0.2097 | 493.13 | 22.636 | 0.4135 | 0.6838 | 18.131 | 0.2263 |
| CFLD (Lu et al., 2024) | 118.10 | 29.030 | 0.3696 | 0.6841 | 7.8766 | 0.2506 | 129.30 | 21.050 | 0.4001 | 0.6269 | 9.5606 | 0.2461 |
| *EgoWorld* (Ours) | **90.893** | **31.004** | **0.4096** | **0.6519** | **7.4087** | **0.2585** | **96.429** | **24.851** | **0.4605** | **0.6188** | **8.1031** | **0.2582** |

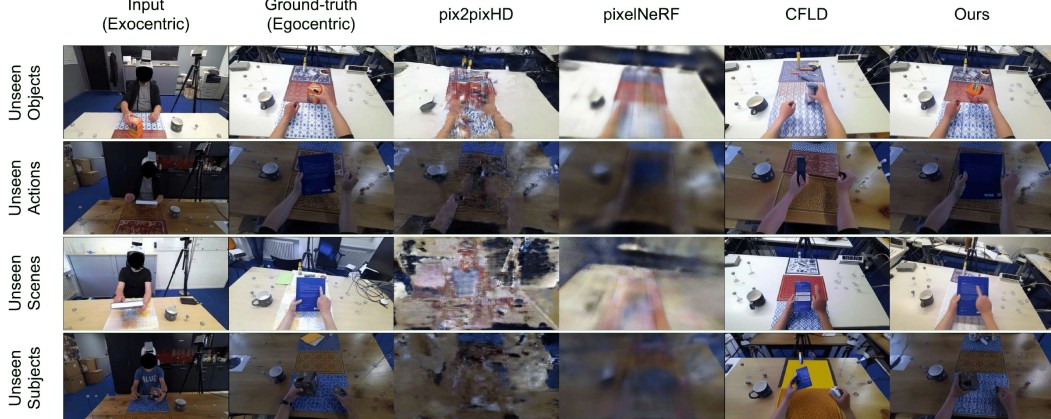

Figure 3: **Comparisons with state-of-the-arts on unseen scenarios (*i.e.,* objects, actions, scenes, and subjects) in H2O (Kwon et al., 2021).** Compared to state-of-the-arts (*i.e.,* pix2pixHD (Wang et al., 2018), pixelNeRF (Yu et al., 2021), and CFLD (Lu et al., 2024)), *EgoWorld* outperforms the image reconstruction quality with respect to hand-object interaction and background regions for all unseen scenarios.

# 4 EXPERIMENTS

## 4.1 DATASETS

To evaluate exocentric-to-egocentric translation models including *EgoWorld*, we select **H2O** (Kwon et al., 2021), which contains diverse scenarios such as unseen objects, actions, scenes, and subjects. Following previous work (Luo et al., 2024b), we split four unseen settings to evaluate generalization as follows: (1) **unseen objects**, where we train with six objects and test with novel two objects, (2) **unseen actions**, where we train with first 80% frames and test with last 20% frames, (3) **unseen scenes**, where we train with four scenes and test with novel two scenes, and (4) **unseen subjects**, where we train with one subject and test with novel one subject. To further demonstrate the generalizability of our method, we also evaluate it on **TACO** (Liu et al., 2024b), **Assembly101** (Sener et al., 2022), and **Ego-Exo4D** (Grauman et al., 2024) datasets. Since they provide hand-object interaction sequences involving 15, 1,380, and 689 actions respectively, we adopt them as **unseen actions** scenario, which allows for a general and comprehensive evaluation of generalization performance.

## 4.2 EVALUATION METRICS

Following previous works (Luo et al., 2024b; Liu et al., 2024a), we adopt a comprehensive set of evaluation metrics to assess reconstruction quality and generalization: (1) **Fréchet Inception Distance (FID)** (Heusel et al., 2017), which uses Inception-v3 (Salimans et al., 2016) features to measure the distributional distance between generated and real images; (2) **Peak Signal-to-Noise Ratio (PSNR)**, a pixel-wise fidelity metric that quantifies the ratio between the maximum possible

Table 2: **Comparisons with state-of-the-arts on unseen actions in TACO (Liu et al., 2024b), Assembly101 (Sener et al., 2022), and Ego-Exo4D (Grauman et al., 2024).** Compared to state-of-the-arts (*i.e.,* pix2pixHD (Wang et al., 2018), pixelNeRF (Yu et al., 2021), and CFLD (Lu et al., 2024)), *EgoWorld* outperforms for all unseen scenarios in all metrics (*i.e.,* FID, PSNR, SSIM, LPIPS, PA-MPJPE, and CLIPScore).

| Methods | FID↓ | PSNR↑ | SSIM↑ | LPIPS↓ | PA-MPJPE↓ | CLIPScore↑ |
|---|---|---|---|---|---|---|
| **TACO (Liu et al., 2024b)** | | | | | | |
| pix2pixHD (Wang et al., 2018) | 227.87 | 25.875 | 0.2806 | 0.7037 | 19.054 | 0.2309 |
| pixelNeRF (Yu et al., 2021) | 302.19 | 26.661 | 0.3888 | 0.8543 | 16.137 | 0.2251 |
| CFLD (Lu et al., 2024) | 61.357 | 28.769 | 0.4009 | 0.5033 | 7.9078 | 0.2715 |
| *EgoWorld* (Ours) | **37.191** | **30.155** | **0.4237** | **0.4025** | **7.3590** | **0.2828** |
| **Assembly101 (Sener et al., 2022)** | | | | | | |
| pix2pixHD (Wang et al., 2018) | 350.97 | 17.107 | 0.3587 | 0.6578 | 21.967 | 0.2114 |
| pixelNeRF (Yu et al., 2021) | 356.44 | 19.037 | 0.3761 | 0.6019 | 19.658 | 0.2070 |
| CFLD (Lu et al., 2024) | 53.931 | 20.998 | 0.3988 | 0.5566 | 11.108 | 0.2458 |
| *EgoWorld* (Ours) | **50.232** | **25.365** | **0.4101** | **0.5142** | **10.561** | **0.2558** |
| **Ego-Exo4D (Grauman et al., 2024)** | | | | | | |
| pix2pixHD (Wang et al., 2018) | 401.48 | 14.792 | 0.3065 | 0.6899 | 25.082 | 0.2203 |
| pixelNeRF (Yu et al., 2021) | 367.39 | 17.347 | 0.3618 | 0.7134 | 23.793 | 0.2149 |
| CFLD (Lu et al., 2024) | 70.476 | 21.578 | 0.3614 | 0.5975 | 15.010 | 0.2670 |
| *EgoWorld* (Ours) | **61.231** | **24.985** | **0.3986** | **0.5482** | **13.992** | **0.2862** |

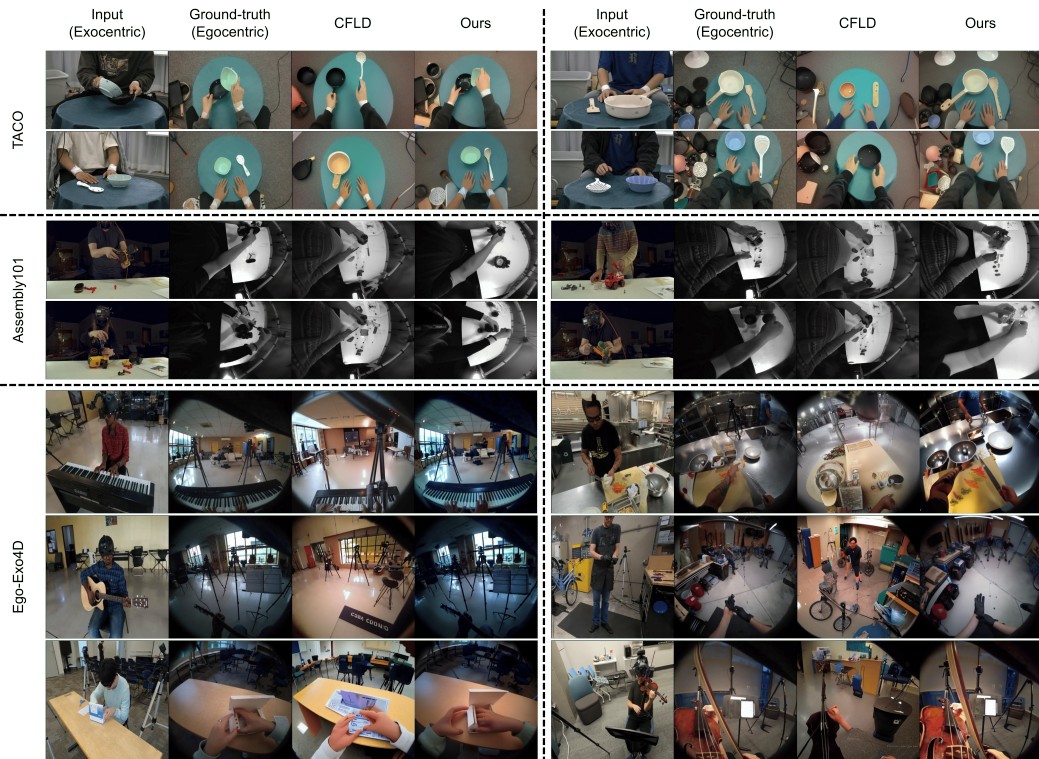

Figure 4: **Comparisons with state-of-the-art on unseen actions scenario in TACO (Liu et al., 2024b), Assembly101 (Sener et al., 2022), and Ego-Exo4D (Grauman et al., 2024).** Compared to state-of-the-art (*i.e.,* CFLD (Lu et al., 2024)), *EgoWorld* outperforms the image reconstruction quality with respect to hand-object interaction and background regions even on more challenging scenarios than H2O (Kwon et al., 2021).

pixel value and the mean squared error (MSE) between a reconstructed image and its ground-truth counterpart; (3) **Structural Similarity Index Measure (SSIM)** (Wang et al., 2004), which evaluates image similarity by comparing luminance, contrast, and structural information to better reflect human visual perception; (4) **Learned Perceptual Image Patch Similarity (LPIPS)** (Zhang et al., 2018), which employs a deep neural network (Simonyan & Zisserman, 2014) trained on human

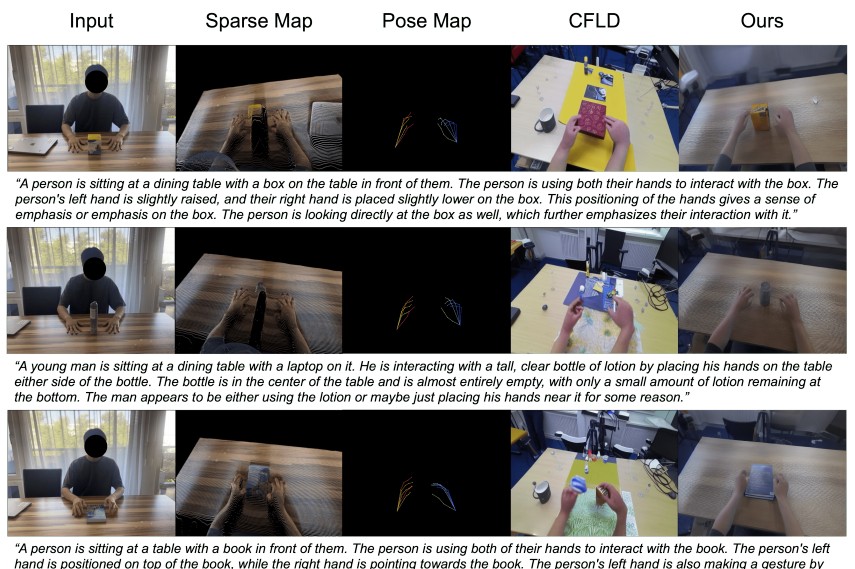

Figure 5: **Real-world comparisons with state-of-the-art.** Compared to state-of-the-art (*i.e.,* CFLD (Lu et al., 2024)), *EgoWorld* significantly outperforms with respect to hand-object interaction and background regions for in-the-wild scenarios.

judgments to assess perceptual similarity; (5) **Procrustes Analysis Mean Per Joint Position Error (PA-MPJPE)**, which measures the average Euclidean distance between predicted and ground-truth 3D hand joints after Procrustes alignment (*i.e.,* scale, rotation, and translation normalization) to evaluate hand generation accuracy, where the predicted 3D hand joints are obtained using HaMeR (Pavlakos et al., 2024); and (6) **CLIPScore** (Hessel et al., 2021), which computes the similarity between image and text embeddings obtained from CLIP (Radford et al., 2021) to assess object-level generalization.

### 4.3 RESULTS

#### 4.3.1 COMPARISONS ON BENCHMARKS

To compare *EgoWorld* with related works, we consider several state-of-the-arts: (1) **pix2pixHD** (Wang et al., 2018), a single-view images-to-image translation model, (2) **pixelNeRF** (Yu et al., 2021), a generalizable neural rendering method that synthesizes novel views from one or few images by combining pixel-aligned features with NeRF-style volume rendering, and (3) **CFLD** (Lu et al., 2024), a coarse-to-fine latent diffusion framework that decouples pose and appearance information at different stages of the generation process. Due to absence of source code of Exo2Ego (Luo et al., 2024b), which estimates egocentric hand layout and generated egocentric image based on the hand layout, we adopt CFLD. Since CFLD assumes ground-truth hand layouts as input, it is an upper-bound reference for Exo2Ego. In addition, since the source code of 4Diff (Cheng et al., 2024), which generates images using only point clouds without hand poses or textual descriptions, is not available, we simulate it by removing pose and text from our model. Experimental results of 4Diff setting can be found in Tab. 3.

Based on experiments on H2O across four unseen scenarios, our method achieves state-of-the-art performance on all evaluation metrics as shown in Tab. 1. pix2pixHD and pixelNeRF perform substantially worse across scenarios, while CFLD serves as the strongest baseline. Compared to CFLD, *EgoWorld* achieves consistent improvements in all unseen settings. On unseen objects, FID is reduced from 59.615 to 41.334 (30% relative reduction), with PSNR improving by over 5 dB (25.922 to 31.171). A similar trend is observed for unseen actions, where FID decreases by 35% (50.953 to 33.284) and PSNR increases by more than 3 dB. Even in the more challenging unseen scene setting, requiring accurate global context reconstruction, *EgoWorld* reduces FID by 23% (118.10 to 90.893).

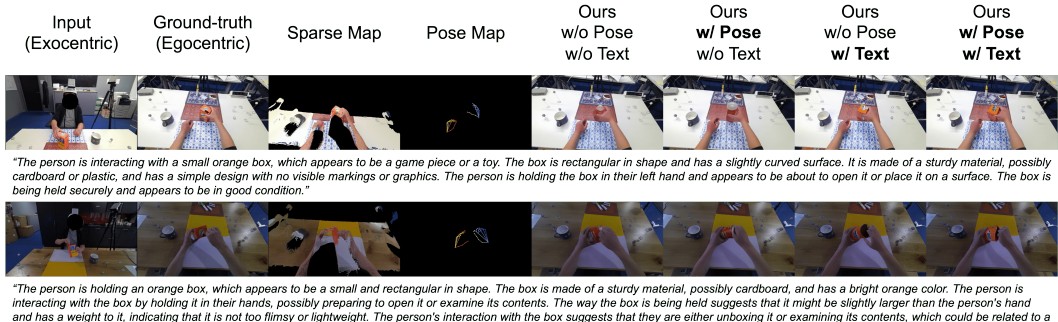

Figure 6: **Ablation study for conditioning modalities.** *EgoWorld* generates more reasonable images when conditioned on both pose maps and text, compared to using only one or none.

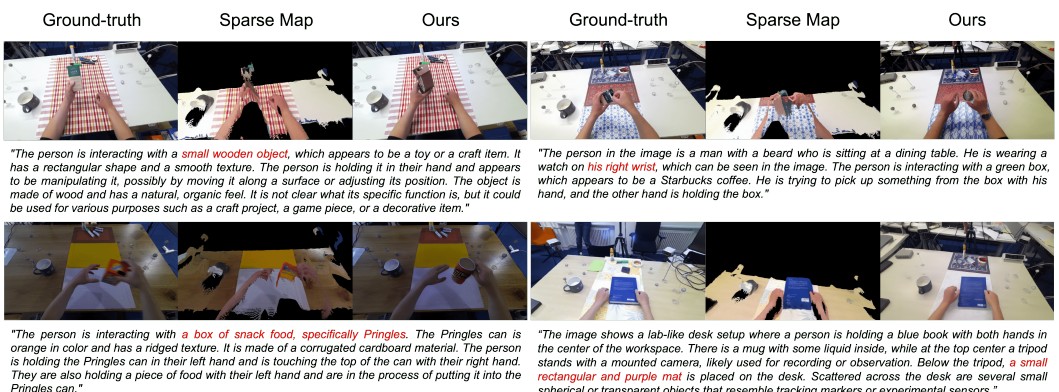

Figure 7: **Ablation study for incorrect textual description.** The red-colored texts represent incorrect descriptions, which are reflected as conditioning inputs for *EgoWorld* to generate egocentric images.

Although the gain in PA-MPJPE is moderate (e.g., 7.9971 to 7.3178 on unseen objects), substantial improvements in FID and LPIPS (0.4539 to 0.3476) indicate enhanced perceptual realism beyond pose alignment. The consistent increase in CLIPScore further suggests improved semantic consistency between generated egocentric views and underlying interactions.

As shown in Fig. 3, pix2pixHD produces noisy artifacts, while pixelNeRF generates blurry outputs lacking fine-grained details. pix2pixHD is not well-suited for exocentric-to-egocentric translation due to large geometric discrepancies, and pixelNeRF is primarily designed for multi-view synthesis. Although CFLD reconstructs hands effectively, it struggles with detailed object appearance and global scene context, resulting in unrealistic backgrounds. Therefoe, *EgoWorld* achieves robust performance even in challenging unseen scenarios, by leveraging complementary cues from the exocentric view, including pose, text, and sparse maps.

Moreover, as shown in Tab. 2 and Fig. 4, *EgoWorld* generalizes effectively to other datasets with increasing real-world complexity. On TACO, our method reduces FID by approximately 39% compared to CFLD (61.357 to 37.191) and improves PSNR by 1.4 dB. On Assembly101, although the absolute FID margin is smaller (53.931 to 50.232), *EgoWorld* consistently outperforms CFLD across all metrics, including a 4.3 dB improvement in PSNR. On Ego-Exo4D, which exhibits substantial real-world variability, our method reduces FID by 13% and improves PSNR by over 3 dB while also lowering PA-MPJPE by more than 1 mm. Across all datasets and unseen settings, these consistent improvements demonstrate that *EgoWorld* effectively reconstructs both local hand details and global scene structure, achieving strong perceptual fidelity, semantic alignment, and pose accuracy in diverse cross-view generation scenarios.

### 4.3.2 COMPARISONS ON REAL-WORLD EXAMPLES

Furthermore, to evaluate real-world generalization in in-the-wild settings, we conduct experiments on *EgoWorld* using a state-of-the-art baseline model for comparison. We collect in-the-wild images of people interacting with arbitrary objects using their hands. *Note that our method relies solely on a single RGB image captured using a smartphone (iPhone 13 Pro), and we apply the complete pipeline without any additional inputs.* As shown in Fig. 5, CFLD produces egocentric images that appear unnatural and biased toward patterns observed in the training data, resulting in inconsistencies when applied to novel interaction scenarios. In contrast, *EgoWorld* generates coherent and realistic egocentric views by effectively leveraging the sparse structural map, demonstrating strong generalization to unseen real-world examples. These results suggest that *EgoWorld* maintains robust performance even in unconstrained environments. With further training on more diverse datasets, the proposed framework has the potential to support practical real-world applications.

### 4.3.3 ABLATION STUDY FOR CONDITIONING MODALITIES

To analyze the contribution of each conditioning modality, we perform an ablation study by selectively enabling pose and text inputs, as shown in Tab. 3. Without pose or text, the model achieves an FID of 56.120 and PSNR of 27.054. Adding pose alone yields only marginal improvement (FID 55.016), whereas incorporating text alone substantially reduces FID to 44.240 (21% relative reduction) and improves PSNR to 28.565, highlighting the im-

Table 3: **Ablation study for conditioning modalities.** *EgoWorld* achieves higher scores when conditioned on both pose maps and text, compared to using only one or none.

| Pose | Text | FID↓ | PSNR↑ | SSIM↑ | LPIPS↓ | PA-MPJPE↓ | CLIPScore↑ |
|---|---|---|---|---|---|---|---|
| | | 56.120 | 27.054 | 0.4460 | 0.4454 | 7.8022 | 0.2713 |
| ✓ | | 55.016 | 27.544 | 0.4449 | 0.4122 | 7.8007 | 0.2720 |
| | ✓ | 44.240 | 28.565 | 0.4573 | 0.3821 | 7.7452 | 0.2729 |
| ✓ | ✓ | **41.334** | **31.171** | **0.4814** | **0.3476** | **7.3178** | **0.2731** |

portance of semantic cues for object and scene reconstruction. The best performance is achieved when both pose and text are jointly used, further reducing FID to 41.334 and increasing PSNR to 31.171 (+2.6 dB over text-only), while also lowering PA-MPJPE to 7.3178. As shown in Fig. 6, removing text leads to incorrect object reconstruction, whereas pose guidance improves hand configuration realism, demonstrating complementary roles of semantic (text) and structural (pose) conditioning. We additionally simulate the 4Diff (Cheng et al., 2024) setting by removing both pose and text conditions (first row), which results in clear performance degradation, indicating that realistic egocentric reconstruction requires both semantic and geometric guidance.

### 4.3.4 ABLATION STUDY FOR INCORRECT TEXTUAL DESCRIPTION

To evaluate the influence of textual guidance on egocentric reconstruction, we intentionally provide textual descriptions that partially mismatch the exocentric image. As shown in Fig. 7, the textual input modulates appearance-level attributes of objects, subjects, and the overall scene in the reconstructed egocentric view. Importantly, despite the semantic mismatch, *EgoWorld* consistently preserves the underlying geometric structure (e.g., table slope) encoded in the sparse map. This behavior indicates that textual guidance affects semantic and appearance components, while the structural layout remains grounded in geometric observations. These results demonstrate that *EgoWorld* can flexibly integrate multi-modal cues, enabling controllable semantic modulation without compromising geometric consistency, even under previously unseen combinations of visual and textual inputs.

## 5 CONCLUSION

In this work, we introduce *EgoWorld*, a novel framework that translates exocentric observations into egocentric views by leveraging rich multi-modal cues. Our two-stage design first extracts informative exocentric observations and then reconstructs realistic egocentric images from sparse egocentric maps through a diffusion model conditioned on pose and text. Through extensive experiments on four datasets (*i.e.,* H2O, TACO, Assembly101, and Ego-Exo4D), we demonstrate that *EgoWorld* outperforms existing methods and proves highly effective. Beyond benchmark performance, *EgoWorld* exhibits strong generalization to real-world samples, highlighting its potential for deployment in diverse and unconstrained scenarios.

## 6 ACKNOWLEDGMENT

This research is funded by an SNSF Postdoc.Mobility Fellowship P500PT_225450.

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
