# A APPENDIX

## A.1 IMPLEMENTATION DETAILS

### A.1.1 EGOCENTRIC VIEW RECONSTRUCTION

To train the egocentric view reconstruction, we fine-tune a pre-trained LDM inpainting model (Rombach et al., 2022). Based on the PyTorch Lightning framework (Falcon & The PyTorch Lightning team, 2019), we set the training settings included a batch size of 3, a learning rate of $1 \times 10^{-5}$, and the AdamW optimizer (Loshchilov & Hutter, 2017), for a total of 5 epochs (about 10 hours). All experiments are conducted on a single NVIDIA RTX 4090 GPU.

### A.1.2 3D EGOCENTRIC HAND POSE ESTIMATOR

To train a 3D egocentric hand pose estimator from exocentric inputs, we adopt a backbone as ViT-224 (Dosovitskiy et al., 2020) and a regressor as MLP, which consists of two linear layers and one ReLU (Nair & Hinton, 2010) between linear layers. The input and output feature dimensions of the first linear layer are 768 and 512, and those of the last linear layer are 512 and 126. Based on the PyTorch framework (Paszke et al., 2019), we set the training settings included a batch size of 64, a learning rate of $1 \times 10^{-4}$, a criterion of MSE loss, and the Adam optimizer (Kingma & Ba, 2015), for a total of 100 epochs (about 20 hours). All experiments were conducted on a single NVIDIA RTX 4090 GPU.

## A.2 MORE RESULTS

Table A: **Comparisons with image completion backbones.** Compared to image completion backbones (*i.e.,* MAE (He et al., 2022) and MAT (Li et al., 2022)), LDM (Rombach et al., 2022) outperforms in all metrics.

| Backbones | FID↓ | PSNR↑ | SSIM↑ | LPIPS↓ | PA-MPJPE↓ | CLIPScore↑ |
|---|---|---|---|---|---|---|
| MAE (He et al., 2022) | 169.91 | 24.623 | 0.4148 | 0.5041 | 10.978 | 0.2564 |
| MAT (Li et al., 2022) | 89.933 | 28.922 | 0.4370 | 0.4758 | 9.5442 | 0.2677 |
| MAT (Refined) (Li et al., 2022) | 68.628 | 29.750 | 0.4731 | 0.4506 | 8.2561 | 0.2603 |
| LDM (Rombach et al., 2022) | **41.334** | **31.171** | **0.4814** | **0.3476** | **7.3178** | **0.2731** |

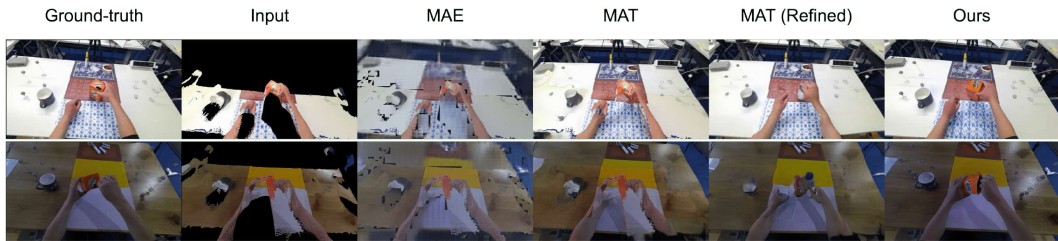

Figure A: **Comparisons with image completion backbones.** Compared to image completion backbones (*i.e.,* MAE (He et al., 2022) and MAT (Li et al., 2022)), LDM (Rombach et al., 2022) outperforms with respect to hand-object interaction and background regions for all cases.

### A.2.1 COMPARISONS WITH IMAGE COMPLETION BACKBONES

To validate the architecture for egocentric view reconstruction, we compare our method with state-of-the-art image completion backbones, including MAE (He et al., 2022), MAT (Li et al., 2022), and LDM (Rombach et al., 2022). MAE focuses on mask-based image encoding for missing region reconstruction, while MAT leverages transformer-based long-range context modeling to restore large masked areas. LDM, which serves as the backbone of *EgoWorld*, differs in its ability to condition on multiple modalities such as text and pose. As shown in Fig. A, the LDM-based method produces more natural and coherent egocentric reconstructions than alternative backbones. Although vanilla MAT effectively fills missing regions, it often introduces contextual inconsistencies (e.g.,

Table B: **Quantitative analysis of pose modeling strategies.** The proposed 3D egocentric hand pose estimator showcases a higher score than other baselines of pose estimation.

| Methods | FID↓ | PSNR↑ | SSIM↑ | LPIPS↓ | PA-MPJPE↓ | CLIPScore↑ |
|---|---|---|---|---|---|---|
| Egocentric Body Pose Estimation | 86.542 | 25.133 | 0.4686 | 0.5365 | 15.897 | 0.2310 |
| Egocentric Camera Pose Estimation | 44.907 | 27.821 | 0.4311 | 0.4809 | 8.0193 | 0.2700 |
| Egocentric Hand Pose Estimation (CNN-based) | 61.162 | 26.034 | 0.4033 | 0.5172 | 10.895 | 0.2620 |
| Egocentric Hand Pose Estimation (ViT-based) (Ours) | **42.323** | **28.897** | **0.4408** | **0.4590** | **7.9645** | **0.2714** |

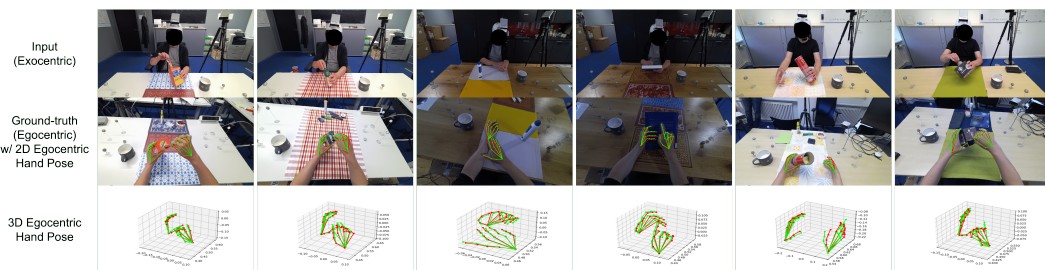

Figure B: **Visual analysis of 3D egocentric hand pose estimator.** Green and red poses indicate the ground-truth and estimated pose, respectively. Estimated poses are well-aligned with the ground-truth both in 2D and 3D spaces.

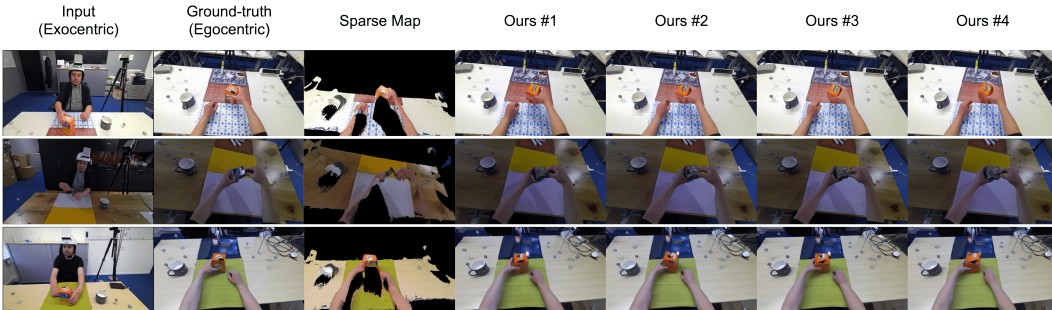

Figure C: **Visual analysis of generation consistency of egocentric view reconstruction.** With four iterations, the outputs are consistent, reliable, and similar to ground-truth.

subtle color discrepancies). We further refine MAT with random patch masking and recovery, which improves contextual blending but still fails to preserve fine-grained hand-object interactions due to limited semantic conditioning. In contrast, the LDM-based approach performs iterative latent denoising with multimodal conditioning, enabling coherent restoration across both local interaction regions and globally consistent areas. Quantitative results in Table A show that our method consistently outperforms all alternatives across evaluation metrics. Based on these findings, we adopt LDM as the backbone architecture for *EgoWorld*.

### A.2.2 ANALYSIS OF POSE MODELING STRATEGIES

To demonstrate the effectiveness of modeling hand poses, we compare our proposed exocentric image-based 3D egocentric hand pose estimator not only with egocentric camera pose estimation but also with whole-body pose estimation approaches. As shown in Tab. B, our hand pose estimation model achieves the best performance among all pose configurations. We further evaluate off-the-shelf whole-body pose estimation models (e.g., Hand4Whole (Moon et al., 2022) and OSX (Lin et al., 2023)), and observe that their performance is consistently lower than that of dedicated hand pose estimation,

Table C: **Comparisons with whole-body and hand pose estimation.** The case of hand pose estimation showcases a higher score than that of whole-body pose estimation.

| Methods | MPJPE ↓ | |
|---|---|---|
| | Left Hand | Right Hand |
| Whole-Body Pose Estimation | 19.52 | 19.49 |
| Hand Pose Estimation (Ours) | **1.005** | **1.161** |

as reported in Tab. C. In exocentric hand-object interaction scenarios, the person is frequently oc-

Table D: **Analysis of MANO and keypoint representations.** The representation of the hand pose does not have a significant impact on performance.

| Representations | FID↓ | PSNR↑ | SSIM↑ | LPIPS↓ | PA-MPJPE↓ | CLIPScore↑ |
|---|---|---|---|---|---|---|
| MANO (Romero et al., 2017) | 33.208 | 31.632 | 0.4609 | 0.3771 | 7.3358 | 0.2812 |
| Keypoint (Ours) | 33.284 | 31.620 | 0.4566 | 0.3780 | 7.2602 | 0.2824 |

Table E: **Analysis of robustness on noisy input.** *EgoWorld* showcases robustness on noisy exocentric input and alleviates the heavy reliance on off-the-shelf estimators.

| Test Sets | Methods | FID↓ | PSNR↑ | SSIM↑ | LPIPS↓ | PA-MPJPE↓ | CLIPScore↑ |
|---|---|---|---|---|---|---|---|
| All Cases | pix2pixHD (Wang et al., 2018) | 211.10 | 24.420 | 0.2854 | 0.6127 | 17.754 | 0.2450 |
| | pixelNeRF (Yu et al., 2021) | 251.76 | 27.061 | 0.3950 | 0.8159 | 14.636 | 0.2315 |
| | CFLD (Lu et al., 2024) | 50.953 | 28.529 | 0.4324 | 0.4593 | 8.1199 | 0.2699 |
| | *EgoWorld* (Ours) | **33.284** | **31.620** | **0.4566** | **0.3780** | **7.2602** | **0.2824** |
| Noisy Cases | pix2pixHD (Wang et al., 2018) | 233.09 | 23.897 | 0.2612 | 0.6553 | 18.453 | 0.2432 |
| | pixelNeRF (Yu et al., 2021) | 255.10 | 26.352 | 0.3892 | 0.8236 | 15.103 | 0.2269 |
| | CFLD (Lu et al., 2024) | 52.879 | 27.090 | 0.4037 | 0.4701 | 8.3807 | 0.2644 |
| | *EgoWorld* (Ours) | **34.910** | **30.284** | **0.4455** | **0.3835** | **7.3895** | **0.2790** |

cluded by desks or tables, making full-body pose estimation inherently unreliable. In contrast, hands remain relatively visible, resulting in more robust and feasible pose estimation.

To further analyze the impact of backbone architecture, we compare CNN-based (i.e., ResNet50 (He et al., 2016)) and transformer-based (ViT Dosovitskiy et al. (2020)) backbones for egocentric hand pose estimation. While the CNN backbone primarily focuses on local regions, the ViT backbone leverages global contextual information, leading to superior performance. These results indicate that modeling hand poses with a global context-aware architecture is particularly beneficial in exocentric observation settings.

Moreover, we conduct a qualitative evaluation to validate the effectiveness of the proposed estimator. As illustrated in Fig. B, given a single exocentric image, our model predicts 3D hand poses that closely align with the ground truth. This demonstrates that the estimator is highly effective not only for computing the transformation matrix during the exocentric observation stage, but also for initializing the hand pose map in the egocentric view reconstruction stage.

Overall, since the hand is the most visible and reliably observable body part in exocentric hand-object interaction scenarios, egocentric hand pose estimation proves to be the most effective strategy, and incorporating a ViT backbone further enhances performance.

### A.2.3 GENERATION CONSISTENCY OF EGOCENTRIC VIEW RECONSTRUCTION

To evaluate the consistency of our generative model, we generated egocentric images multiple times under identical conditions. As shown in Fig. C, we present four outputs generated from the same exocentric image and corresponding sparse map, and our model consistently produces coherent egocentric images across runs. Despite the inherent variability in generative models, our method achieves stable and reliable exocentric-to-egocentric view translation, demonstrating its robustness and consistency.

### A.2.4 EFFECT OF HAND REPRESENTATION

To examine the effect of MANO (Romero et al., 2017) representation for hand pose, we build an egocentric MANO parameter estimator based on ViT (Dosovitskiy et al., 2020) and MLP layers, and validate final results on the egocentric view reconstruction stage. As shown in Tab. D, the trivial difference of performance on MANO is revealed. Although MANO representation contains richer visual information than keypoints, it does not exert a strong influence in the egocentric view reconstruction stage, as hand pose is fused with other modalities, *i.e.,* sparse maps and text descriptions.

Table F: **Analysis of individual sub-modules of exocentric view observation.** Whether using the ground-truth or not, *EgoWorld* outperforms baselines which use the ground-truth. Underlined results indicate the case that no ground-truths were provided.

| Methods | Pose | Depth | Text | FID↓ | PSNR↑ | SSIM↑ | LPIPS↓ | PA-MPJPE↓ | CLIPScore↑ |
|---|---|---|---|---|---|---|---|---|---|
| pix2pixHD (Wang et al., 2018) | GT | – | – | 211.10 | 24.420 | 0.2854 | 0.6127 | 17.754 | 0.2450 |
| pixelNeRF (Yu et al., 2021) | GT (Camera) | – | – | 251.76 | 27.061 | 0.3950 | 0.8159 | 14.636 | 0.2315 |
| CFLD (Lu et al., 2024) | GT | – | – | 50.953 | 28.529 | 0.4324 | 0.4593 | 8.1199 | 0.2699 |
| *EgoWorld* (Ours) | Prediction | Prediction | Prediction (Gemini Team et al. (2023)) | 42.323 | 28.897 | 0.4408 | 0.4590 | 7.9645 | 0.2714 |
|  | Prediction | GT | Prediction (Qwen-VL Bai et al. (2023)) | 41.198 | 29.002 | 0.4420 | 0.4379 | 7.9074 | 0.2740 |
|  | GT | Prediction | Prediction (Qwen-VL Bai et al. (2023)) | 37.040 | 30.017 | 0.4487 | 0.4092 | 7.8256 | 0.2761 |
|  | GT | GT | Prediction (Gemini Team et al. (2023)) | 34.891 | 30.998 | 0.4501 | 0.3820 | 7.4909 | 0.2790 |
|  | GT | GT | Prediction (Qwen-VL Bai et al. (2023)) | **33.284** | **31.620** | **0.4566** | **0.3780** | **7.2602** | **0.2824** |

Table G: **Impact of the depth estimator and 3D egocentric hand pose estimator.** *EgoWorld* outperforms baselines that do not use these components.

| Methods | FID↓ | PSNR↑ | SSIM↑ | LPIPS↓ | PA-MPJPE↓ | CLIPScore↑ |
|---|---|---|---|---|---|---|
| w/o Depth Estimator | 71.461 | 26.807 | 0.3961 | 0.7013 | 14.032 | 0.2468 |
| w/o 3D Egocentric Hand Pose Estimator | 62.714 | 27.002 | 0.4071 | 0.5121 | 8.5976 | 0.2557 |
| *EgoWorld* (Ours) | **42.323** | **28.897** | **0.4408** | **0.4590** | **7.9645** | **0.2714** |

Table H: **Results of the video-to-video extension framework.** While the video-to-video extension framework improves temporal consistency, it leads to a decrease in image quality.

| Methods | T-LPIPS↑ | Flow-warp↑ | FID↓ | PSNR↑ | SSIM↑ | LPIPS↓ | PA-MPJPE↓ | CLIPScore↑ |
|---|---|---|---|---|---|---|---|---|
| w/o Video Extension (Ours) | 0.9827 | 0.9792 | **33.284** | **31.620** | **0.4566** | **0.3780** | **7.2602** | **0.2824** |
| w/ Video Extension | **0.9860** | **0.9827** | 35.455 | 30.279 | 0.4409 | 0.3791 | 7.3461 | 0.2805 |

### A.2.5 ROBUSTNESS ON NOISY INPUT

With our proposed pipeline, the heavy reliance on off-the-shelf estimators is likely to create error propagation vulnerabilities under occlusion or noisy inputs. Thus, we conduct additional experiments on how much the noisy input affects the final result. We newly define a noisy test set from H2O (Kwon et al., 2021) unseen actions scenario, which contains the cases causing incorrect depth or hand pose estimation (*e.g.,* occluded hands by object or hand, or blurry hand). We manually select hard cases. As shown in Tab. E, there was a slight deterioration in performance for the noisy cases, but it still achieved outstanding performance compared to other baselines. Although the off-the-shelf estimators may introduce some noise or slightly lower accuracy, our model demonstrates significantly greater robustness compared to other baselines. This indicates that even with current state-of-the-art estimators, our framework can produce reliable results. We expect even better performance in the future as estimation models continue to improve.

### A.2.6 IMPACT OF SUB-MODULES OF EXOCENTRIC VIEW OBSERVATION

To evaluate the impact of individual sub-modules (*i.e.*, hand pose estimator, depth estimator, and vision-language model (VLM)) in the observation pipeline, we conduct an experiment on H2O (Kwon et al., 2021) unseen actions scenario by distinguishing whether each sub-module is used or the ground-truth is used. Note that since there are no ground-truths for text description in the H2O dataset, we quantify the impact of VLM by comparing Qwen-VL (Bai et al., 2023), which we already adopted, with Gemini (Team et al., 2023), which is the popular foundation model. As shown in Tab. F, all prediction cases (fourth row) record the lowest score for all metrics. However, this case outperforms all state-of-the-art baselines, which use ground-truth hand pose or camera pose. It implies although the performance of each sub-module is crucial, we expect the improvement of sub-modules will further increase our framework's performance in the future.

Furthermore, to conduct a more thorough test on sub-modules, we conducted additional experiments: (1) We removed the depth estimator and examined whether egocentric view reconstruction is still feasible using only the exocentric image instead of the sparse map. (2) We removed the 3D egocentric hand pose estimator and investigated whether the model can still reconstruct the egocentric view using only the exocentric hand pose map instead of the egocentric hand pose map. As shown in Tab. G, performance degradation was observed across all metrics. These results confirm that both the sparse map derived from the depth estimator and the egocentric hand pose obtained from the egocentric hand pose estimator are essential for accurate egocentric view reconstruction.

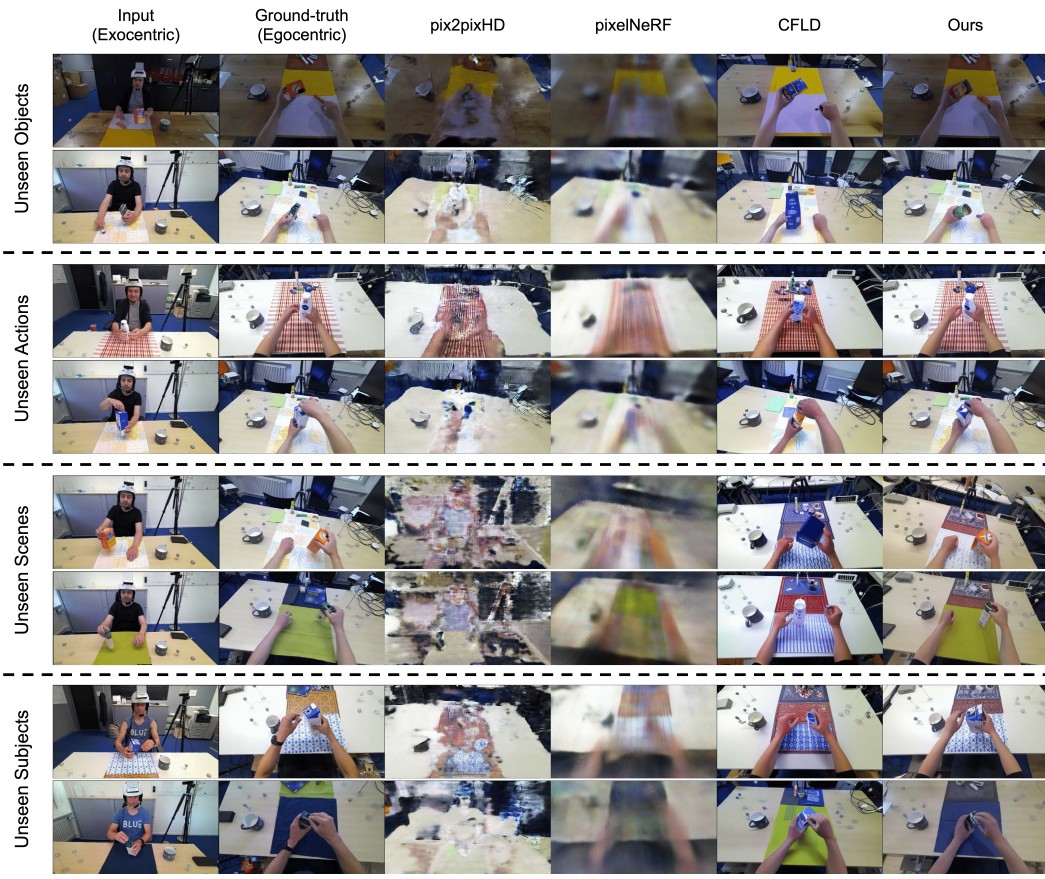

Figure D: **Additional comparisons with state-of-the-arts on unseen scenarios.** Compared to state-of-the-arts (*i.e.,* pix2pixHD (Wang et al., 2018), pixelNeRF (Yu et al., 2021), and CFLD (Lu et al., 2024))), *EgoWorld* outperforms for all unseen scenarios.

### A.2.7 EXTENSION TO VIDEO-TO-VIDEO TRANSLATION FRAMEWORK

To explore the potential extension to a video-based framework, we implemented a mechanism to partially incorporate the latent embedding of the previous frame when generating the next frame. Specifically, the first frame is generated from random noise, and for subsequent frames, the latent embedding is constructed by combining the previously generated latent embedding with new random noise. The combination ratio is set to latent embedding : random noise = 1 : 9. This choice is intentional; if the latent embedding dominates, the differences between consecutive frames diminish, resulting in nearly static frames. By emphasizing random noise, we preserve temporal variation and enable dynamic frame generation. Furthermore, to evaluate temporal consistency, we adopted two metrics: T-LPIPS and Flow-warp. T-LPIPS measures temporal consistency by computing the perceptual distance (LPIPS) between consecutive frames. We calculate $1 - LPIPS$ so that higher scores indicate smoother transitions and fewer perceptual fluctuations. Flow-warp evaluates temporal stability by estimating optical flow to warp the previous frame toward the next frame and measuring the difference. A higher score indicates that motion and appearance remain consistent over time, reflecting stronger temporal coherence. Therefore, as shown in Tab. H, T-LPIPS and Flow-warp are improved compared to before applying this mechanism. However, in terms of egocentric view reconstruction, image-level quality metrics show marginal drops. This trade-off is commonly observed in video generation and is consistent with prior works. In future work, we plan to explore temporal consistency further by incorporating temporal layers, as proposed in AnimateDiff (Guo et al., 2023).

| Input (Exocentric) | Ground-truth (Egocentric) | Ours | | Input (Exocentric) | Ground-truth (Egocentric) | Ours |

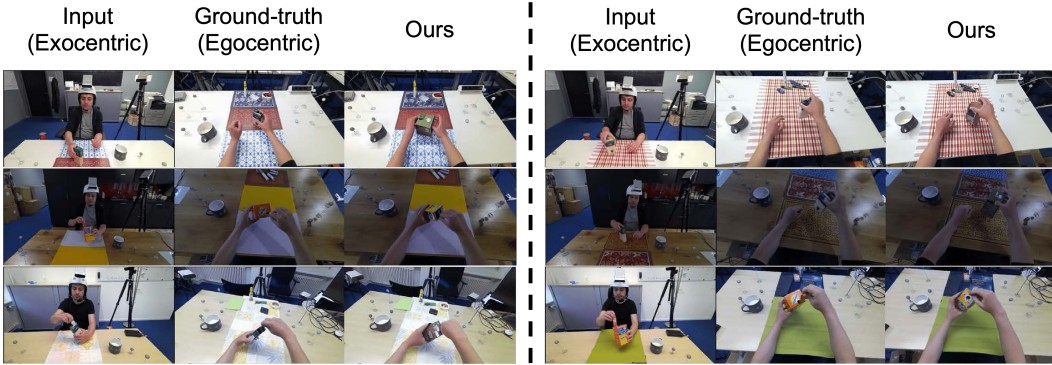

Figure E: **Failure examples.** Subtle finger movements and dependency of VLMs make the reconstructed ouptuts of hands and objects quite unsatisfying.

### A.2.8 Additional Comparisons with State-of-the-Arts

We provide additional state-of-the-art comparisons on H2O (Kwon et al., 2021) as shown in Fig. D. We evaluate our method across four unseen scenarios (*i.e.,* unseen objects, actions, scenes, and subjects) and observe that it consistently outperforms baseline models. pix2pixHD (Wang et al., 2018), which depends on label map-based image-to-image translation, generates egocentric images with significant noise; it implies pix2pixHD is ill-suited for tackling the exocentric-to-egocentric view translation task. Likewise, pixelNeRF (Yu et al., 2021), which is originally intended for novel view synthesis using multiple inputs, produces blurry results that lack fine-grained details; it means pixelNeRF is less effective for one-to-one view translation. On the other hand, CFLD (Lu et al., 2024), which focuses on generating view-aware person images using hand pose maps, shows better performance than the previous methods. However, its strengths are largely confined to hand region translation only, and it struggles to accurately reconstruct surrounding information like objects and scenes. In contrast, our approach, *EgoWorld*, produces robust and coherent results even in complex and previously unseen scenarios involving rich contextual elements. Therefore, we verify *EgoWorld*'s generalization ability across diverse, unseen situations.

### A.3 Limitations and Future Work

In Fig. E, we present representative failure cases on H2O (Kwon et al., 2021). In certain examples, the reconstructed hand poses or manipulated objects deviate from the ground-truth. For hand poses, subtle finger articulations that are barely observable in the exocentric view remain challenging to infer accurately in the egocentric reconstruction. This limitation suggests the need for more robust 3D egocentric hand pose estimation, potentially through temporally consistent modeling, uncertainty-aware pose regression, or tighter integration between pose and depth estimation to produce more reliable sparse maps for hand-aligned reconstruction. For object reconstruction, regions that are heavily occluded or entirely invisible in the exocentric image may result in distorted or implausible reconstructions in the egocentric view. Moreover, inaccuracies in text descriptions generated by VLMs from exocentric observations can propagate errors to the final reconstruction.

Future work could explore stronger cross-modal alignment mechanisms, joint optimization of visual and textual representations, or the incorporation of geometry-aware priors to improve robustness against incomplete observations. We anticipate that advances in multi-modal reasoning and vision-language modeling will further enhance reconstruction fidelity in such challenging scenarios.

## B LLM Usage

Large language models (LLMs) were used solely for language editing and writing assistance. Specifically, they were used to improve the clarity, grammar, and general readability of the manuscript. The LLMs did not contribute to research ideation, experimental design, implementation, or analysis. All technical content, results, and conclusions are solely the responsibility of the authors.