# OpenReview forum: "EgoWorld: Translating Exocentric View to Egocentric View using Rich Exocentric Observations"
_ICLR.cc/2026/Conference — ICLR 2026 Poster_

### Official Review · Reviewer_jkpb · 2025-10-30

**Soundness:** 3
**Presentation:** 3
**Contribution:** 3
**Rating:** 6
**Confidence:** 4

**Summary:**

This paper introduces EgoWorld, a two-stage framework to translate a single exocentric image into a high-fidelity egocentric view. Stage one extracts geometric and semantic cues (a sparse reprojected point cloud, 3D egocentric hand pose, and text) from the exocentric image. Stage two uses a diffusion model to inpaint and reconstruct the final egocentric view, conditioned on these cues. The method shows great performance across four datasets.

**Strengths:**

- The paper addresses the challenging task of translating third-person to first-person views, a problem with clear applications in AR/VR and robotics.
- The method design makes a lot of sense to me. It logically separates 3D geometric reprojection (creating a sparse map) from generative inpainting (using diffusion with multimodal cues) to reconstruct the final image.
- EgoWorld achieves strong results on four different datasets, with thorough testing against unseen objects, actions, scenes, and subjects, including in-the-wild examples. The provided ablation study is helpful.

**Weaknesses:**

- I feel the method's success critically depends on the 3D egocentric hand pose estimator ($\phi_{ego}$), which predicts the *egocentric* hand pose from the *exocentric* image. This key component is only briefly described in the main paper. It's unclear what insights make this difficult cross-view prediction possible. More importantly, the paper should justify why it uses hand poses ($P_{exo}$ and $P_{ego}$) for the Umeyama alignment instead of simply predicting the egocentric *head pose*. Using the head pose seems like a much more direct way to find the camera's transformation matrix, and the benefit of the hand-based alignment is not explained.
- The proposed method is image-based, but the paper's motivation, applications, and datasets are all video-centric. It would be helpful if the authors discuss how this single-frame approach could be extended to a full video-to-video translation framework. A discussion on the expected challenges, such as maintaining temporal consistency across generated frames, would be a valuable addition.
- The name of the first stage, "Exocentric View Observation" is confusing. This stage *outputs* an "egocentric RGB map $S_{ego}$" and a "3D egocentric hand pose $P_{ego}$". A name like "Exocentric-to-Egocentric Cue Translation" would be more accurate and less confusing.

**Questions:**

See weaknesses for questions.

---

> ### Author Response · Authors · 2025-11-21
> **Rebuttal for Reviewer jkpb**
>
> **Q1: Dependency on 3D Egocentric Hand Pose Estimator**
>
> **A1:**
> Thank you for the insightful comment. We emphasize the 3D egocentric hand pose estimator for the following reasons:
>
> (1) In exocentric views, hand-object interactions are generally visible, making it intuitive to predict egocentric hand poses from these cues. In contrast, the human head may be occluded in certain cases, making head pose estimation less intuitive and often infeasible. For example, in the TACO and Assembly101 datasets, the human head is not always visible, rendering head pose estimation unreliable. However, the human hand is consistently visible, making it a more effective cue.
>
> (2) The key to accurate exocentric-to-egocentric alignment lies in resolving depth ambiguity, which is crucial for reconstructing the egocentric view in metric space. Similar to prior work [1], we leverage the 3D egocentric hand pose, which can be estimated in metric scale, to perform Umeyama alignment. Using the hand pose rather than the head pose provides a reliable geometric anchor for this transformation, since hand joints offer well-defined, spatially distributed keypoints that are visible and trackable across views.
>
> (3) As shown in Tab. 3 of the revised paper, incorporating egocentric hand poses as a condition for egocentric view reconstruction improves plausible hand generation. Using head poses instead would remove egocentric hand information, making it difficult to generate natural-looking hands. This is further supported by the introduction of PA-MPJPE, a metric specifically designed to evaluate hand generation accuracy, where our method achieves significantly better performance.
>
> **Q2: Extension to Video-to-Video Translation Framework**
>
> **A2:**
> Thank you for the valuable suggestion. To explore the potential extension to a video-based framework, we implemented a mechanism to partially incorporate the latent embedding of the previous frame when generating the next frame. Specifically, the first frame is generated from random noise, and for subsequent frames, the latent embedding is constructed by combining the previously generated latent embedding with new random noise.
> The combination ratio is set to latent embedding : random noise = 1 : 9. This choice is intentional; if the latent embedding dominates, the differences between consecutive frames diminish, resulting in nearly static frames. By emphasizing random noise, we preserve temporal variation and enable dynamic frame generation.
>
> Furthermore, to evaluate temporal consistency, we adopted two metrics: T-LPIPS and Flow-warp.
> T-LPIPS measures temporal consistency by computing the perceptual distance (LPIPS) between consecutive frames. We calculate 1 - LPIPS so that higher scores indicate smoother transitions and fewer perceptual fluctuations.
> Flow-warp evaluates temporal stability by estimating optical flow to warp the previous frame toward the next frame and measuring the difference. A higher score indicates that motion and appearance remain consistent over time, reflecting stronger temporal coherence.
> Therefor, as shown in the table below, T-LPIPS and Flow-warp are improved compared to before applying this mechanism. However, in terms of egocentric view reconstruction, image-level quality metrics show marginal drops. This trade-off is commonly observed in video generation and is consistent with prior works.
> In future work, we plan to explore temporal consistency further by incorporating temporal layers, as proposed in AnimateDiff [2].
>
> | Methods               | T-LPIPS↑ | Flow-warp↑ | FID↓     | PSNR↑   | SSIM↑   | LPIPS↓  | PA-MPJPE↓ | CLIPScore↑ |
> |----------------------|-----------|------------|----------|---------|---------|---------|-----------|------------|
> | w/o Video Extension   | 0.9827   | 0.9792     | **33.284** | **31.620** | **0.4566** | **0.3780** | **7.2602** | **0.2824** |
> | w/ Video Extension    | **0.9860** | **0.9827** | 35.455  | 30.279  | 0.4409  | 0.3791  | 7.3461    | 0.2805     |

---

> ### Author Response · Authors · 2025-11-21
> **Rebuttal for Reviewer jkpb**
>
> **Q3: Confusion of Name of First Stage**
>
> **A3:**
> In our proposed “Exocentric View Observation,” the model outputs an egocentric RGB sparse map and a 3D egocentric hand pose as reviewer pointed out. However, prior to this, it also outputs an exocentric textual description, a 3D exocentric hand pose, and an exocentric depth map. Furthermore, the actual exocentric-to-egocentric translation occurs during “Egocentric View Reconstruction,” so the reviewer’s suggested term, “Exocentric-to-Egocentric Cue Translation,” could be interpreted as encompassing the entire framework. We are open to considering alternative terminology that more precisely reflects this process.
>
> > [1] Changwoon Choi, Jeongjun Kim, Geonho Cha, Minkwan Kim, Dongyoon Wee, and Young Min Kim. Humans as a calibration pattern: Dynamic 3d scene reconstruction from unsynchronized and uncalibrated videos. In ICCV, pp. 6598–6608, 2025.
> >
> > [2] Yuwei Guo, Ceyuan Yang, Anyi Rao, Zhengyang Liang, Yaohui Wang, Yu Qiao, Maneesh Agrawala, Dahua Lin, and Bo Dai. Animatediff: Animate your personalized text-to-image diffusion models without specific tuning. arXiv preprint arXiv:2307.04725, 2023.

---

### Official Review · Reviewer_u9sS · 2025-11-01

**Soundness:** 2
**Presentation:** 3
**Contribution:** 2
**Rating:** 4
**Confidence:** 4

**Summary:**

This paper proposes EgoWorld, a two-staged framework that generates an egocentric view from external camera views. EgoWorld extracts multimodal cues, such as textual descriptions, hand pose, and depth map from external views. Then, the egocentric view is estimated sparsely from the pointcloud generated from depth map and relative transform between egocentric and exocentric view. This is further refined using predicted hand pose and a textual description via a latent diffusion model to produce an egocentric image. Method is evaluated on H2O, TACO, Assembly101, and Ego-Exo4D and achieves state-of-the-art performance.

**Strengths:**

* Framework utilizes multi-modal information, such as textual description, from an exocentric view in a plausible way.
* The overall proposed method is well presented with clarity.

**Weaknesses:**

* Evaluation compares the proposed framework to novel view synthesis methods, many of which are outdated and not specialized for the egocentric setup. This is particularly problematic when there are methods that focus on an egocentric view.
* Egocentric view transform is estimated based on the egocentric hand pose predicted from the external view. This would be highly dependent on the learned camera setup and the device's visual from an external view. Using ground-truth data in some evaluation further weakens the claim that such a framework is advantageous.

**Questions:**

* Some experiments in the appendix should instead be in the main paper (e.g., results using prediction instead of GT for H2O).
* Previous exocentric-to-egocentric translation approaches can be evaluated with the same setup and framework, or partially ablated to use the same input (one camera, single frame) to demonstrate the advantage of the proposed framework.
* In particular, Exo2Ego's hand-object interaction layout requirement doesn't significantly differ from using hand pose in the proposed method. They can be easily compared by providing a hand-only layout estimated in the same way as the proposed method.
* The evaluation should show whether using hand pose for relative camera pose estimation is more advantageous than a wider variety of baselines (despite one naive comparison in the appendix). The claim of being camera parameter-free is not significant without it.

---

> ### Author Response · Authors · 2025-11-21
> **Rebuttal for Reviewer u9sS**
>
> **Q1: Evaluation of Exocentric-to-Egocentric Translation Approaches with Same Setup and Framework**
>
> **A1:**
> Direct comparison with egocentric-specific models such as Exo2Ego and 4Diff is challenging due to the unavailability of their code.
>
> (1) Therefore, we compared our method with CFLD, a model that generates target-view images based on hand layouts similar to Exo2Ego. Notably, unlike Exo2Ego, CFLD assumes ground-truth hand layouts as input, which makes it a much stronger and more informative baseline. In other words, CFLD can be viewed as an upper-bound reference for Exo2Ego, as it removes errors from hand-pose estimation and provides the model with perfect pose supervision. Consequently, CFLD is expected to outperform Exo2Ego in image generation quality, and serves as a more rigorous and competitive baseline for evaluating our approach.
>
> (2) To approximate the 4Diff setting, which utilizes only point clouds without additional cues, we conducted an ablation in which hand-pose and textual inputs were removed from our model, as reported in Tab. 3. We note that this is not intended to claim a one-to-one reproduction of 4Diff, since the architectural design and overall modeling philosophy differ. Instead, this comparison is meant to provide a controlled perspective on how comprehensive exocentric observation with multi-modalities contribute within our framework, and to show that the improvements do not arise from a simple combination of modalities but from how these cues are integrated and utilized during generation.
>
> As shown in the table below, *EgoWorld* achieves the best performance. These results demonstrate the effectiveness of actively leveraging multiple modalities—pose, text, and point cloud—for egocentric view generation.
>
> | Methods             | FID↓     | PSNR↑   | SSIM↑   | LPIPS↓  | PA-MPJPE↓ | CLIPScore↑ |
> |--------------------|----------|---------|---------|---------|-----------|------------|
> | Exo2Ego-setting     | 59.615  | 25.922  | 0.4307  | 0.4539  | 7.9971    | 0.2656     |
> | 4Diff-setting       | 56.120  | 27.054  | 0.4460  | 0.4454  | 7.8022    | 0.2713     |
> | *EgoWorld (Ours)* | **41.334** | **31.171** | **0.4814** | **0.3476** | **7.3178** | **0.2731** |
>
> **Q2: Dependency on Learned Camera and Device from External View**
>
> **A2:**
> We fully agree with the reviewer’s comment regarding ground-truth dependency. To address this, we conducted experiments on H2O unseen actions scenario, reported in Tab. F of the supplementary material and the table below, where hand poses were replaced with predicted results instead of ground-truth. While performance dropped compared to using ground-truth poses, *EgoWorld* still outperformed other baseline models that relied on ground-truth poses.
> This demonstrates that even when pose quality decreases, *EgoWorld*’s strategy of leveraging other modalities effectively compensates for egocentric image generation, minimizing dependency on external views. These results from the appendix have been moved to the main paper in the updated version.
>
> | Methods               | Pose       | FID↓     | PSNR↑   | SSIM↑   | LPIPS↓  | PA-MPJPE↓ | CLIPScore↑ |
> |----------------------|-----------|----------|---------|---------|---------|-----------|------------|
> | pix2pixHD             | GT        | 211.10  | 24.420  | 0.2854  | 0.6127  | 17.754    | 0.2450     |
> | pixelNeRF             | GT        | 251.76  | 27.061  | 0.3950  | 0.8159  | 14.636    | 0.2315     |
> | CFLD                  | GT        | 50.953  | 28.529  | 0.4324  | 0.4593  | 8.1199    | 0.2699     |
> | *EgoWorld (Ours)*     | Prediction| 41.198  | 29.002  | 0.4420  | 0.4379  | 7.9074    | 0.2740     |
> | *EgoWorld (Ours)*   | GT        | **33.284** | **31.620** | **0.4566** | **0.3780** | **7.2602** | **0.2824** |
>
> **Q3: Advantage of Hand Pose for Relative Camera Pose Estimation**
>
> **A3:**
> To demonstrate the effectiveness of hand poses, we compared our proposed egocentric hand pose estimation model not only with the previously tested egocentric camera pose estimation but also with egocentric body pose estimation. The results show that our hand pose estimation model achieves the best performance. In particular, since the person is often occluded by tables or other objects in the exocentric view, body pose estimation is inherently prone to infeasible results.
> Additionally, to examine the impact of backbone architecture, we compared CNN (i.e., ResNet50) and ViT backbones for egocentric hand pose estimation. The results indicate that, unlike the CNN backbone, which focuses on local regions, the ViT backbone can leverage global and wider contextual information, leading to superior performance.
> In conclusion, the most visible body part in the exocentric view is the hand, making egocentric hand pose estimation the most effective. Furthermore, modeling it with a ViT backbone helps achieve higher performance.

---

> ### Author Response · Authors · 2025-11-21
> **Rebuttal for Reviewer u9sS**
>
> **Q3: Advantage of Hand Pose for Relative Camera Pose Estimation (continued)**
>
> | Methods                                    | FID↓     | PSNR↑   | SSIM↑   | LPIPS↓  | PA-MPJPE↓ | CLIPScore↑ |
> |-------------------------------------------|----------|---------|---------|---------|-----------|------------|
> | Egocentric Body Pose Estimation            | 86.542  | 25.133  | 0.4686  | 0.5365  | 15.897    | 0.2310     |
> | Egocentric Camera Pose Estimation          | 44.907  | 27.821  | 0.4311  | 0.4809  | 8.0193    | 0.2700     |
> | Egocentric Hand Pose Estimation (CNN-based)| 61.162  | 26.034  | 0.4033  | 0.5172  | 10.895    | 0.2620     |
> | Egocentric Hand Pose Estimation (ViT-based) (Ours) | **42.323** | **28.897** | **0.4408** | **0.4590** | **7.9645** | **0.2714** |

---

### Official Review · Reviewer_pmAo · 2025-11-01

**Soundness:** 3
**Presentation:** 2
**Contribution:** 3
**Rating:** 4
**Confidence:** 3

**Summary:**

The paper proposes a new framework, EgoWorld, for translating third-person views into first-person (ego) view images. Unlike prior methods that depend on multi-view inputs or known ego-camera poses, EgoWorld introduces a two-stage framework. It first extracts multimodal information—such as 3D hand pose, textual description, and depth—from a single third-person image using existing specialized models. These modalities are then integrated into a diffusion-based image reconstruction model to generate the first-person view. Experiments on four public datasets (H2O, TACO, Assembly101, and Ego-Exo4D) demonstrate that EgoWorld achieves state-of-the-art performance and shows promising generalization to unseen scenarios.

**Strengths:**

- The concept of using textual cues to enhance visual synthesis across viewpoints is intuitive and promising.

- The combination of sparse maps as partial observation with textual description for completion is interesting.

**Weaknesses:**

1. It would be better to highlight the difference compared with 4Diff or Exo2Ego. 4Diff adopts depth maps/point clouds while Exo2Ego adopts hand layout. In addition to depth maps and hand poses, the proposed method adopts an additional textual description for LDM as a condition, which like a combination.

2. The introduction of textual cues is the key of the proposed method and provides semantic alignment. However, the quality of the text and its contribution to the results is unclear. It would be better to include metrics like clip score to measure the alignment between text and generated images. Moreover, in Fig.C, only the incorrect textual description for hand-held objects is shown, which is insufficient. Most importantly, it would be interesting to see the balance between sparse maps and incorrect textual description like Fig.D. Also, those analysis are important and should in the main paper.

3. The comparison with baselines seems unfair. For example, CFLD is primarily designed for pose-guided person image synthesis in general human view translation, not specifically for hand-centric first-person generation. It would be better to incorporate more directly relevant comparisons (e.g., Exo2Ego or EgoExo-Gen).

4. It would be better to provide qualitative results and more analysis for unseen objects or novel environments and to show its strong generalization capability. Based on L.81-84 ("Yet, it depends heavily...overfitting to the training dataset"), overfitting is easy to happen and can be alleviated by exocentric observations. However, Fig.4 shows that the images generated by the proposed method are close to the groundtruth even in unknown regions, which also raises the concerns about overfitting.

**Questions:**

See Weaknesses.

---

> ### Author Response · Authors · 2025-11-21
> **Rebuttal for Reviewer pmAo**
>
> **Q1: Difference from 4Diff and Exo2Ego**
>
> **A1:**
> Thank you for the insightful comment. Although our method shares the use of depth maps and point clouds with 4Diff, the key difference lies in the transformation strategy. 4Diff assumes a predefined transformation matrix to convert an exocentric point cloud into its egocentric counterpart, which limits its applicability in in-the-wild scenarios where such calibration is not available. In contrast, *EgoWorld* predicts this transformation from paired exocentric–egocentric hand poses, allowing the system to operate reliably and flexibly in realistic settings without requiring manual scene alignment.
>
> Similarly, while our method also incorporates hand layouts like Exo2Ego, relying solely on hand structure for viewpoint translation imposes strong constraints. With only a hand layout as input, the model must reconstruct the egocentric view without additional contextual cues such as point clouds or textual descriptions, increasing the risk of overfitting to scene configurations present in the training set. To address this limitation, *EgoWorld* integrates multiple complementary modalities (i.e., 3D geometric cues, hand poses, and textual descriptions), which together provide a richer and more informative representation of the exocentric scene. Although observing exocentric scene is straightforward idea, effectively integrate it in one pipeline is not trivial and it has been positively noted by other reviewers as intuitive (**reviewer pZGe & jkpb**), plausible (**reviewer u9sS**), and effective (**reviewer pZGe**).
>
> Furthermore, *EgoWorld* adopts a logical two-stage decomposition that separates 3D geometric reprojection (used to produce sparse egocentric maps) from generative inpainting using diffusion models conditioned on multimodal cues. This architectural separation enables accurate geometric grounding while allowing flexible and expressive visual synthesis during the final reconstruction. As highlighted by **reviewers pmAo & u9sS**, this pipeline leverages exocentric observations in a plausible and well-justified manner, leading to reconstructions that better generalize to previously unseen environments. With these design choices, *EgoWorld* effectively overcomes the limitations of both 4Diff and Exo2Ego. Although they do not open source code, we experimentally simulate them to compare with our model as follows:
>
> (1) To simulate the 4Diff setting, which generates images using only point clouds without hand poses or textual descriptions, we conducted experiments in which pose and text inputs were removed from our model, as reported in Tab. 3.
>
> (2) We compared our method with CFLD, a model that generates target-view images based on hand layouts similar to Exo2Ego. Notably, unlike Exo2Ego, CFLD assumes ground-truth hand layouts as input, making it an upper-bound reference for Exo2Ego. Consequently, CFLD is expected to outperform Exo2Ego in image generation quality.
>
> As shown in the table below, *EgoWorld* achieves the best performance. These results demonstrate the effectiveness of actively leveraging multiple modalities—pose, text, and point cloud—for egocentric view generation.
>
> | Methods             | FID↓     | PSNR↑   | SSIM↑   | LPIPS↓  | PA-MPJPE↓ | CLIPScore↑ |
> |--------------------|----------|---------|---------|---------|-----------|------------|
> | Exo2Ego-setting     | 59.615  | 25.922  | 0.4307  | 0.4539  | 7.9971    | 0.2656     |
> | 4Diff-setting       | 56.120  | 27.054  | 0.4460  | 0.4454  | 7.8022    | 0.2713     |
> | *EgoWorld (Ours)* | **41.334** | **31.171** | **0.4814** | **0.3476** | **7.3178** | **0.2731** |
>
> **Q2: Evaluation for Semantic Alignment**
>
> **A2:**
> We appreciate the reviewer’s suggestion. To evaluate the alignment between the textual description and the generated image, we adopted CLIPScore [1] as a metric. As shown in the table below, *EgoWorld* demonstrates significantly stronger text–image alignment compared to other baseline models. This trend is consistently observed not only on H2O, but also across TACO, Assembly101, and Ego-Exo4D. Therefore, *EgoWorld* maintains robust semantic alignment performance across all scenarios.

---

> ### Author Response · Authors · 2025-11-21
> **Rebuttal for Reviewer pmAo**
>
> **Q2: Evaluation for Semantic Alignment (continued)**
>
> | Methods                   | H2O - Unseen Objects | H2O - Unseen Actions | H2O - Unseen Scenes | H2O - Unseen Subjects | TACO   | Assembly101 | Ego-Exo4D |
> |----------------------------|--------------------|--------------------|--------------------|----------------------|--------|-------------|------------|
> | pix2pixHD     | 0.2302             | 0.2450             | 0.2159             | 0.2311               | 0.2309 | 0.2114      | 0.2203     |
> | pixelNeRF         | 0.2270             | 0.2315             | 0.2097             | 0.2263               | 0.2251 | 0.2070      | 0.2149     |
> | CFLD              | 0.2656             | 0.2699             | 0.2506             | 0.2461               | 0.2715 | 0.2458      | 0.2670     |
> | *EgoWorld (Ours)*      | **0.2731**         | **0.2824**         | **0.2585**         | **0.2582**           | **0.2828** | **0.2558** | **0.2862** |
>
> Additionally, to complement the results that regarding incorrect textual descriptions for hand-held objects, we further examined the impact of inaccurate textual descriptions and sparse maps from the perspectives of scene and subject. As shown in Fig. 7 of the revised paper, we observed that textual descriptions are capable of altering not only the object in the egocentric view but also the appearance, shape, and color of the subject and the scene itself. Importantly, even when provided with inaccurate textual descriptions, *EgoWorld* consistently preserves the geometric structure (e.g., slope of table) encoded in the sparse map. This expanded analysis has been incorporated into the main paper.
>
> **Q3: Fair Comparison**
>
> **A3:**
> While CFLD is indeed a model designed for pose-guided person image synthesis, we selected it as a baseline for the following reasons:
>
> (1) CFLD is structurally a pose-map-based transfer model, which makes it agnostic to whether the pose is a full-body human pose or a hand pose. In other words, when trained from scratch using hand pose maps on hand-centric datasets (i.e., H2O, TACO, Assembly101, and Ego-Exo4D), CFLD can serve as a hand-centric first-person image generation model.
>
> (2) CFLD with groud-truth hand poses serves as an upper-bound reference for Exo2Ego. While Exo2Ego predicts hand pose maps, CFLD assumes the hand pose maps are given as ground-truth. Therefore, CFLD is expected to outperform Exo2Ego in image generation quality.
>
> (3) The code for directly related models such as Exo2Ego and EgoExo-Gen is not publicly available. CFLD was selected as the baseline that could be reimplemented most faithfully within our framework and experimental constraints, allowing for the closest possible comparison under a consistent setup.
>
> Additionally, EgoExo-Gen is a first-frame-based future egocentric video generation model, making direct comparison unfair. Our model, in contrast, does not require the first frame and can generate the egocentric view purely from exocentric observations, addressing a fundamentally more challenging problem than EgoExo-Gen. Moreover, the problem formulation itself is fundamentally different: EgoExo-Gen assumes that the target egocentric viewpoint is already available at the first frame and focuses on predicting subsequent frames, whereas our method tackles exocentric-to-egocentric translation, where the egocentric view is not given at all. Therefore, the two methods operate under different assumptions and address different tasks, and a direct quantitative comparison is not fully aligned.
>
> **Q4: More Qualitative Analysis for Unseen Objects or Novel Environments**
>
> **A4:**
> As mentioned in Section 4.1, Fig. 4 presents the results of testing unseen action scenarios on TACO, Assembly101, and Ego-Exo4D. However, as the reviewer pointed out, these results may not clearly demonstrate generalization to unseen objects or novel environments. In particular, since all scenarios in TACO and Assembly101 were captured in constrained laboratory settings, the results may appear overfitted.
>
> To address this, we conducted unseen scenario experiments on in-the-wild examples as shown in Fig. 5 of the revised paper.
> We take in-the-wild images of people interacting with arbitrary objects using their hands. Note that we rely solely on a single RGB image captured using a smartphone (iPhone 13 Pro) and apply our complete pipeline. No additional information beyond this single exocentric image is used.
> As a result, *EgoWorld* still demonstrates superior generalization compared to the baseline models. This confirms that leveraging multiple modalities—such as point clouds, textual descriptions, and hand poses—effectively aids in generating generalized egocentric images.
>
> > [1] Jack Hessel, Ari Holtzman, Maxwell Forbes, Ronan Le Bras, and Yejin Choi. Clipscore: A reference-free evaluation metric for image captioning. In EMNLP, pp. 7514–7528, 2021.

---

### Official Review · Reviewer_pZGe · 2025-11-03

**Soundness:** 3
**Presentation:** 3
**Contribution:** 3
**Rating:** 6
**Confidence:** 4

**Summary:**

This paper addresses the task of exocentric-to-egocentric cross-view translation. The authors propose a two-stage training framework to tackle this problem: (1) extracting point clouds, 3D hand poses, and textual descriptions from the exocentric view, and (2) reconstructing the egocentric view based on these extracted cues. The key idea is to leverage multimodal information, including textual supervision, 3D geometric cues, and hand pose data to better guide the exo-to-ego synthesis process. The proposed approach is evaluated on four datasets: H2O, TACO, Assembly101, and Ego-Exo4D.

**Strengths:**

- The paper is well-written and easy to follow.

- The idea of leveraging multimodal information—including rich textual supervision, 3D geometric cues, and hand pose data—is simple, intuitive, and effective.

- The evaluation is comprehensive, covering four representative ego–exo datasets.

**Weaknesses:**

- **Qualitative Results**: Most of the qualitative examples are drawn from the H2O dataset except for Fig 4. It would be beneficial to include more examples from the other three datasets—TACO, Assembly101, and Ego-Exo4D—to showcase a broader range of scenarios beyond desktop activities. This would help readers gain a more comprehensive understanding of the proposed approach’s generalizability across diverse environments.

- **Ablation Study**: In Table 3, the presented ablation results are somewhat limited. Including a more complete ablation table in the supplementary material would provide a clearer and more comprehensive analysis of the contribution of each component within the framework.

- **Backbone Model**: To further improve reconstruction quality, it may be beneficial to adopt more recent video diffusion models, such as Wan or CogVideoX, as the backbone. The current choice, LDM (published in 2022), appears somewhat outdated compared to recent advancements in video generation.

- **Quantitative Evaluation**: The quantitative evaluation primarily focuses on image-based quality metrics. It would strengthen the analysis to include additional metrics assessing hand generation accuracy and object-level generalization, providing a more holistic evaluation of the model’s performance.

**Questions:**

Please see the weaknesses.

---

> ### Author Response · Authors · 2025-11-21
> **Rebuttal for Reviewer pZGe**
>
> **Q1: Qualitative Results**
>
> **A1:**
> Thank you for your thoughtful feedback. Following your suggestion, we expanded our examples of TACO, Assembly101, and Ego-Exo4D. As shown in Fig. 4 in the revised paper, *EgoWorld* demonstrates strong generalization performance beyond desktop activities, compared to state-of-the-art.
> Moreover, to evaluate the influence of textual guidance on egocentric view reconstruction, we newly added the qualitative ablation study for incorrect textual description in Fig. 7 of the revised paper.
>
> **Q2: Ablation Study**
>
> **A2:**
> To provide a comprehensive analysis of the contribution of each component within the framework, we conducted ablation studies as follows: (1) backbones of egocentric view reconstruction, (2) egocentric hand pose estimator, and (3) impact of off-the-shelf modules (i.e., exocentric 3D hand pose estimator, depth estimator, and vision-language model).
>
> (1) As shown in Tab. A in the appendix and the table below, we found LDM is the most powerful backbone for integrating multimodal information (i.e., textual description, pose map, and sparse map).
>
> | Backbones          | FID↓     | PSNR↑   | SSIM↑   | LPIPS↓  | PA-MPJPE↓ | CLIPScore↑ |
> |-------------------|----------|---------|---------|---------|-----------|------------|
> | MAE        | 169.91  | 24.623  | 0.4148  | 0.5041  | 10.978    | 0.2564     |
> | MAT        | 89.933  | 28.922  | 0.4370  | 0.4758  | 9.5442    | 0.2677     |
> | MAT (Refined) | 68.628  | 29.750  | 0.4731  | 0.4506  | 8.2561    | 0.2603     |
> | LDM | **41.334** | **31.171** | **0.4814** | **0.3476** | **7.3178** | **0.2731** |
>
> (2) In addition, as depicted in Tab. B in the appendix and the table below, we prove our ViT-based egocentric hand pose estimator is superior to various baselines, such as CNN-based model, body pose estimator, and camera pose estimator.
>
> | Methods                                           | FID↓     | PSNR↑   | SSIM↑   | LPIPS↓  | PA-MPJPE↓ | CLIPScore↑ |
> |--------------------------------------------------|----------|---------|---------|---------|-----------|------------|
> | Egocentric Body Pose Estimation                  | 86.542  | 25.133  | 0.4686  | 0.5365  | 15.897    | 0.2310     |
> | Egocentric Camera Pose Estimation                | 44.907  | 27.821  | 0.4311  | 0.4809  | 8.0193    | 0.2700     |
> | Egocentric Hand Pose Estimation (CNN-based)     | 61.162  | 26.034  | 0.4033  | 0.5172  | 10.895    | 0.2620     |
> | Egocentric Hand Pose Estimation (ViT-based) (Ours) | **42.323** | **28.897** | **0.4408** | **0.4590** | **7.9645** | **0.2714** |
>
> (3) Moreover, as demonstrated in Tab. F in the appendix and the table below, we investigated the impact of sub-modules by using ground-truth or not, and showcased superior results compared to ground-truth-based state-of-the-arts.
>
> | Methods                   | Pose       | Depth      | Text                                      | FID↓     | PSNR↑   | SSIM↑   | LPIPS↓  | PA-MPJPE↓ | CLIPScore↑ |
> |----------------------------|------------|------------|-------------------------------------------|----------|---------|---------|---------|-----------|------------|
> | pix2pixHD       | GT         | --         | --                                        | 211.10  | 24.420  | 0.2854  | 0.6127  | 17.754    | 0.2450     |
> | pixelNeRF          | GT (Camera)| --         | --                                        | 251.76  | 27.061  | 0.3950  | 0.8159  | 14.636    | 0.2315     |
> | CFLD               | GT         | --         | --                                        | 50.953  | 28.529  | 0.4324  | 0.4593  | 8.1199    | 0.2699     |
> | *EgoWorld* (Ours)         | Prediction | Prediction | Prediction (Gemini )         | 42.323  | 28.897  | 0.4408  | 0.4590  | 7.9645    | 0.2714     |
> |  *EgoWorld* (Ours)        | Prediction | GT         | Prediction (Qwen-VL)           | 41.198  | 29.002  | 0.4420  | 0.4379  | 7.9074    | 0.2740     |
> |  *EgoWorld* (Ours)  | GT         | Prediction | Prediction (Qwen-VL)           | 37.040  | 30.017  | 0.4487  | 0.4092  | 7.8256    | 0.2761     |
> |   *EgoWorld* (Ours)  | GT         | GT         | Prediction (Gemini)         | 34.891  | 30.998  | 0.4501  | 0.3820  | 7.4909    | 0.2790     |
> | *EgoWorld* (Ours)   | GT         | GT         | Prediction (Qwen-VL)           | **33.284** | **31.620** | **0.4566** | **0.3780** | **7.2602** | **0.2824** |

---

> ### Author Response · Authors · 2025-11-21
> **Rebuttal for Reviewer pZGe**
>
> **Q2: Ablation Study (continued)**
>
> Furthermore, we conducted more thorough ablation studies:
> (4) We removed the depth estimator and examined whether egocentric view reconstruction is still feasible using only the exocentric image instead of the sparse map.
> (5) We removed the 3D egocentric hand pose estimator and investigated whether the model can still reconstruct the egocentric view using only the exocentric hand pose map instead of the egocentric hand pose map.
>
> The experiments were conducted under the unseen action scenario of the H2O dataset, and as shown in the table below, performance degradation was observed across all metrics. These results confirm that both the sparse map derived from the depth estimator and the egocentric hand pose obtained from the egocentric hand pose estimator are essential for accurate egocentric view reconstruction. This result has been incorporated into the revised version of the paper.
>
> | Methods                           | FID↓     | PSNR↑   | SSIM↑   | LPIPS↓  | PA-MPJPE↓ | CLIPScore↑ |
> |----------------------------------|----------|---------|---------|---------|-----------|------------|
> | w/o Depth Estimator               | 71.461  | 26.807  | 0.3961  | 0.7013  | 14.032    | 0.2468     |
> | w/o Egocentric Hand Pose Estimator| 62.714  | 27.002  | 0.4071  | 0.5121  | 8.5976    | 0.2557     |
> | *EgoWorld (Ours)*               | **42.323** | **28.897** | **0.4408** | **0.4590** | **7.9645** | **0.2714** |
>
> **Q3: Backbone Model**
>
> **A3:**
> Our method is designed as an image-to-image generation framework rather than a full image-to-video generation model. In contrast, Wan and CogVideoX, which were suggested by the reviewer, are text-to-video generative models and thus differ fundamentally in both purpose and architectural design. These models are not directly aligned with our image-to-image approach.
>
> Moreover, many recent works [1-4] continue to adopt LDM-based pipelines for effective fusion of diverse information sources, demonstrating that LDM remain a strong and versatile modeling choice in current multimodal research. In addition, directly replacing the backbone with more complex DiT-based video diffusion models is not a straightforward modification, as such models require substantial architectural changes and do not necessarily guarantee stable or improved performance in our multimodal setting. Importantly, the core contribution of this paper lies not in introducing a new diffusion backbone, but in proposing a framework for effective multimodal observation fusion and adaptation, which is orthogonal to the choice of the underlying diffusion generator.
>
> **Q4: Quantitative Evaluation**
>
> **A4:**
> Beyond traditional image-quality metrics, we introduced additional evaluation measures that assess hand generation accuracy and object-level generalization.
>
> (1) First, we adopted PA-MPJPE, a metric that measures hand generation accuracy. This allows us to evaluate how well the hand is reconstructed in the generated egocentric image. Specifically, we compute the error between the 3D hand pose of the ground-truth egocentric image and from the generated egocentric image using an off-the-shelf 3D hand pose estimator [5]. As shown in the table below, *EgoWorld* significantly outperforms other comparison models in accurately reconstructing hands in the egocentric view.
>
> (2) In addition, we introduced CLIPScore [6] as a metric to evaluate object-level generalization. This score measures the correlation between the object-level description and the generated image. As shown in the table, *EgoWorld* demonstrates strong object-level generalization, outperforming other comparison models.
>
> These results have been included in the revised version of the paper, and we have consolidated the results for PA-MPJPE and CLIPScore across all tables in the paper.

---

> ### Author Response · Authors · 2025-11-21
> **Rebuttal for Reviewer pZGe**
>
> **Q4: Quantitative Evaluation (continued)**
>
> | Methods                   | PA-MPJPE↓ (H2O - Unseen Objects) | CLIPScore↑ (H2O - Unseen Objects) | PA-MPJPE↓ (H2O - Unseen Actions) | CLIPScore↑ (H2O - Unseen Actions) | PA-MPJPE↓ (H2O - Unseen Scenes) | CLIPScore↑ (H2O - Unseen Scenes) | PA-MPJPE↓ (H2O - Unseen Subjects) | CLIPScore↑ (H2O - Unseen Subjects) |
> |----------------------------|----------------------------|-----------------------------|----------------------------|-----------------------------|---------------------------|----------------------------|-----------------------------|------------------------------|
> | pix2pixHD      | 18.007                     | 0.2302                      | 17.754                     | 0.2450                      | 20.229                    | 0.2159                     | 21.357                      | 0.2311                       |
> | pixelNeRF    | 15.746                     | 0.2270                      | 14.636                     | 0.2315                      | 17.085                    | 0.2097                     | 18.131                      | 0.2263                       |
> | CFLD        | 7.9971                     | 0.2656                      | 8.1199                     | 0.2699                      | 7.8766                    | 0.2506                     | 9.5606                      | 0.2461                       |
> | *EgoWorld (Ours)*      | **7.3178**                 | **0.2731**                  | **7.2602**                 | **0.2824**                  | **7.4087**                | **0.2585**                 | **8.1031**                  | **0.2582**                   |
>
> | Methods                   | PA-MPJPE↓ (TACO) | CLIPScore↑ (TACO) | PA-MPJPE↓ (Assembly101) | CLIPScore↑ (Assembly101) | PA-MPJPE↓ (Ego-Exo4D) | CLIPScore↑ (Ego-Exo4D) |
> |----------------------------|-----------------|------------------|-------------------------|--------------------------|-----------------------|------------------------|
> | pix2pixHD      | 19.054          | 0.2309           | 21.967                  | 0.2114                   | 25.082                | 0.2203                 |
> | pixelNeRF     | 16.137          | 0.2251           | 19.658                  | 0.2070                   | 23.793                | 0.2149                 |
> | CFLD         | 7.9078          | 0.2715           | 11.108                  | 0.2458                   | 15.010                | 0.2670                 |
> | *EgoWorld (Ours)*      | **7.3590**      | **0.2828**       | **10.561**              | **0.2558**               | **13.992**            | **0.2862**             |
>
>
> > [1] Shengqi Liu, Yuhao Cheng, Zhuo Chen, Xingyu Ren, Wenhan Zhu, Lincheng Li, Mengxiao Bi, Xiaokang Yang, and Yichao Yan. Multimodal latent diffusion model for complex sewing pattern generation. In ICCV, pp. 17640–17650, 2025b.
> >
> > [2] Jiaqi Liu, Jichao Zhang, Paolo Rota, and Nicu Sebe. Multi-focal conditioned latent diffusion for person image synthesis. In CVPR, pp. 16019–16028, 2025a.
> >
> > [3] Xinzhe Zhang, Junjie Liang, Peng Cao, Jinzhu Yang, and Osmar R Zaiane. Structure-aware mri translation: Multi-modal latent diffusion model with arbitrary missing modalities. In MICCAI, pp. 508–518, 2025.
> >
> > [4] Junkil Park, Youhan Lee, and Jihan Kim. Multi-modal conditional diffusion model using signed distance functions for metal-organic frameworks generation. Nature Communications, 16(1):34, 2025.
> >
> > [5] Georgios Pavlakos, Dandan Shan, Ilija Radosavovic, Angjoo Kanazawa, David Fouhey, and Jitendra Malik. Reconstructing hands in 3d with transformers. In CVPR, pp. 9826–9836, 2024.
> >
> > [6] Jack Hessel, Ari Holtzman, Maxwell Forbes, Ronan Le Bras, and Yejin Choi. Clipscore: A reference-free evaluation metric for image captioning. In EMNLP, pp. 7514–7528, 2021.

---

### Author Response · Authors · 2025-12-01
**Final Remark of EgoWorld**

We sincerely thank all reviewers for their constructive feedback. Below, we summarize the reviewers’ strengths and concerns, how we addressed them, and the resulting expectations.

## 1. Overview


*EgoWorld* introduces a **new multimodal framework** for translating exocentric to egocentric view by jointly reasoning over **3D geometry, depth, hand pose, and text.**
Through extensive new experiments, ablations, and analyses, we demonstrate that *EgoWorld*:


- **Generalizes** across scenes, objects, and real-world settings,
- **Outperforms** strong baselines and simulated prior frameworks,
- **Remains robust** even with imperfect predicted inputs.


## 2. Reviewers’ Strengths


Across all reviewers, several strengths were consistently highlighted:


- **Strong, intuitive multimodal design** (Reviewer pZGe, pmAo, u9sS): *EgoWorld* effectively leverages **textual supervision, 3D geometric cues, sparse depth/pose maps, and hand pose**, which is an intuitive and promising combination that reviewers found both simple and effective.
- **Comprehensive evaluation** (Reviewer pZGe, jkpb): *EgoWorld* is thoroughly evaluated on **four representative ego–exo datasets**, with extensive tests on **unseen objects, actions, scenes, subjects**, and qualitatively tested even on **in-the-wild cases.**
- **Principled framework structure** (Reviewer jkpb): The separation between **3D geometric reprojection (sparse map construction) and generative inpainting via multimodal diffusion** was repeatedly praised as logical, clean, and well-motivated.
- **Interesting use of text + sparse maps** (Reviewer pmAo): The idea of combining **partial 3D observations** with **textual description for completion** was viewed as novel and promising.
- **Practical impact and relevance** (Reviewer jkpb): Reviewers noted clear applications in **AR/VR and robotics**, and found the use of multimodal information from exocentric views both plausible and impactful.
- **Clear and well-written paper** (Reviewer pZGe, u9sS): Reviewers found the method easy to follow and well-presented.


## 3. Reviewers’ Concerns & How We Fully Addressed Them


Below we summarize all major concerns and how they were resolved:


**A. Need for more qualitative & quantitative evidence** (Reviewer pZGe, pmAo)


- Added extensive qualitative comparisons across TACO, Assembly101, Ego-Exo4D (Fig. 4).
- Added new PA-MPJPE & CLIPScore metrics (Tabs. 1,2), showing significant gains.
- Added robustness analysis with incorrect text prompts (Fig. 7).


**B. Comparison with 4Diff & Exo2Ego** (Reviewer pmAo, u9sS)


- Simulated 4Diff setting (by removing pose/text).
- Compared against CFLD (upper bound for Exo2Ego).
- *EgoWorld* outperforms both, validating multimodal design.


**C. Reliance on hand pose** (Reviewer u9sS, jkpb)


- Added analysis showing hands are the only consistently visible and reliable anchor.
- ViT-based estimator significantly outperforms CNN/body/camera pose baselines (Tab. B).


**D. Fairness of baselines & generalization** (Reviewer pmAo, u9sS)


- Included in-the-wild smartphone results (Fig. 5).
- Demonstrated strong performance under unseen objects/environments.


**E. Additional ablation studies & backbone justification** (Reviewer pZGe)


- New ablations on backbones, predicted vs. GT inputs, and modality removal (Tabs. A,B,F).
- Demonstrated that LDM-based multimodal fusion remains the most stable choice.
- Showed clear necessity of depth + hand pose (Tab. G).

**F. Video extension explanation** (Reviewer jkpb)


- Added latent-blending mechanism and improved temporal metrics (T-LPIPS, Flow-warp) (Tab. H).
- Discussed expected image–video trade-offs and future extensions.


## 4. Rating Context & Expectation


The initial reviewer scores were:


- Reviewer jkpb: 6 (Confidence 4)
- Reviewer pZGe: 6 (Confidence 4)
- Reviewer u9sS: 4 (Confidence 4)
- Reviewer pmAo: 4 (Confidence 3)


Two reviewers already assessed the paper as **above the acceptance threshold (6/10)**, and the remaining two provided a borderline rating while explicitly requesting additional experiments and clarifications.


We have **carefully addressed all concerns** with substantial new experiments, ablations, and analyses. Since the earlier comments primarily requested further evidence, and although reviewers did not have enough time to respond, we hope that **the strengthened experiments and clarifications will allow them to reassess the paper in a positive light.**


## 5. Closing Statement

*EgoWorld* is now significantly stronger:

- **Novel multimodal observation-to-egocentric framework**
- **Extensive new ablations validating comprehensive design choice**
- **Robustness with imperfect inputs**
- **Superior performance to strong and simulated prior baselines**
- **Generalization to real-world, unseen scenarios**

We respectfully request the Area Chair’s favorable consideration based on the strengthened contributions and empirical evidence. Thank you for your time and evaluation.

---

### Meta-Review · Area_Chair_xWmy · 2026-01-02

**Summary:**

This paper introduces EgoWorld, a two-stage framework for translating exocentric observations into egocentric views by jointly leveraging geometry, hand pose, and textual cues. Reviewers consistently recognize the method as well-motivated and technically sound, with a clean decomposition between geometric reprojection and multimodal diffusion-based reconstruction. During the rebuttal, the authors add substantial experimental evidence and analyses that address earlier concerns on baseline fairness, modality contribution, and generalization. After carefully reviewing the revised evidence and discussion, the AC recommends acceptance.

**Reviewer Concerns:**

Reviewer concerns primarily focus on evaluation completeness, comparison to prior exo-to-ego methods, and the reliance on hand pose for alignment. These concerns are addressed through additional quantitative metrics, expanded qualitative results across datasets, controlled ablations isolating each modality, and clearer justification of design choices. The rebuttal also strengthens the discussion on generalization, robustness to imperfect inputs, and practical applicability. No major technical concerns remain outstanding.

**Reviewer Scores:**

- Reviewer pZGe: Initially assigns a score of 6, citing clarity, soundness, and intuitive multimodal design, with requests for additional evidence. Following the rebuttal, the assessment would remain clearly above the acceptance threshold (6).
- Reviewer pmAo: Initially assigns a score of 4, noting questions on comparison fairness and semantic alignment. After the rebuttal addressing these points, the assessment would increase to a positive level (6).
- Reviewer u9sS: Initially assigns a score of 4, expressing concerns about baseline relevance and pose dependence. After the rebuttal and added analyses, the assessment would increase to a positive level (6).
- Reviewer jkpb: Initially assigns a score of 6, recognizing strong design and comprehensive evaluation while raising questions about specific design choices. After the rebuttal, the assessment would remain positive (6).

---

### Decision · Program_Chairs · 2026-01-26

Accept (Poster)